# MODEL UTILITY LAW: EVALUATING LLMS BEYOND PERFORMANCE VIA MECHANISTICALLY INTERPRETABLE METRIC

## ABSTRACT

Conventional performance scores capture only a fraction of a large language model's (LLM) true ability, leaving real-world capability under-evaluated. To close this gap, we advocate a new generalizable evaluation paradigm that estimates a model's unseen capacities — those not covered by testing samples. We introduce the Model Utilization Index (MUI) to realize this paradigm. Inspired by mechanistic interpretability, MUI measures the proportion of neurons or features activated during inference, quantifying the effort a model expends on a task. Our central hypothesis is that a stronger model should achieve higher performance while exhibiting lower utilization, reflecting efficient use of its internal capacity. Experiments across diverse LLMs reveal an inverse relationship between MUI and accuracy, establishing the Utility Law and several corollaries that guide pre-/post-training strategies, enable fairer ranking, and highlight correlations with data diversity. MUI thus provides a scalable, mechanism-aware foundation for evaluating LLMs beyond traditional benchmarks. Our code can be found at https://anonymous.4open.science/r/MUI

## 1 INTRODUCTION

In the era of large language models (LLMs), the conventional evaluation paradigm faces serious challenges. Across different stages of assessment, researchers often find that benchmark scores fail to match real-world experience. We identify two key reasons for this gap. First, LLMs exhibit remarkable generalization ability, capable of handling an almost unlimited range of tasks, while test samples inevitably cover only a narrow slice of that space. As a result, current evaluation methods can capture only part of a model's true capability, creating a mismatch between performance metrics and actual ability. Second, it is no longer feasible to measure generalization by the traditional in-domain/out-of-domain split. Given the massive scale of pre-training data, it is nearly impossible to determine which samples a model has previously seen and which it has not. These limitations call for a new evaluation paradigm — how to estimate a model's capabilities that may not be covered by the given limited testing data, namely generalizable evaluation (Cao et al., 2025).

To this end, we propose a new metric, Model Utilization Index (MUI), designed to complement conventional performance scores and enable a novel generalizable evaluation paradigm. The core idea is analogous to assessing a person's ability through a specific task: we should not only measure the final outcome (i.e., performance metric) but also the amount of effort expended during the process, which we term model utilization. Accordingly, a comprehensive evaluation must consider two dimensions for any given dataset: a truly capable model should 1) expend less effort; and 2) achieve superior performance. The central question, then, is how to quantify the model's effort via MUI?

Inspired by recent advances in mechanistic interpretability, we define MUI as a measure of how fully a model deploys its internal capabilities during inference. Specifically, the proportion of neurons or features activated to complete a given task. This allows MUI general to different interpretability methods. For example, neuron-level analyses assume that a model's knowledge or skills are encoded in specific neurons (Pan et al., 2023). Under this view, the neurons activated during inference represent the "effort" the model expends, regardless of whether the output is correct. In contrast, sparse auto-encoder (SAE) approaches posit that a model's behavior involves all neurons and there-

Figure 1: Illustration of model utility: how much effort utilized to complete given tasks, which can be quantified by Neuron- and SAE-based interpretability, although they have different assumptions.

fore train an auxiliary bypass module in the high-dimensional feature space to disentangle specific capabilities (He et al., 2024b). When the model exhibits a particular skill, only a small subset of these learned features is activated. Therefore, MUI can incorporate either perspective by quantifying the activated neurons or features in their own space. As illustrated in Figure 1, completing a given task typically engages only a fraction of the model's total capacity, and MUI captures this activation ratio as a direct measurement of that effort.

To validate this new evaluation paradigm, we conducted a series of experiments from perspectives:

- Empirical fitting. In a manner analogous to scaling-law analysis, we fit curves for different models in the MUI–performance space. The results consistently reveal an inverse relationship, aligning with our two-dimensional hypothesis mentioned above (less effort yet superior performance). From this we derive the Model Utility Law and Corollaries 1 & 2, which can offer practical guidance for both pre-training and post-training strategies (Section 3.2 3.3 3.4).

- Quantitative comparison with traditional evaluation. We compare model rankings generated by our method against those based solely on conventional performance scores. Ours achieve higher correlation with ground truth and lower variance also implies the stability. This analysis yields Corollary 3 towards a fairer model comparison (Section 3.5).

- Correlation with other metrics. Fixing the model and varying the data, we observe a positive relationship between MUI and established diversity metrics. This leads to Corollary 4, suggesting that MUI can help identify more diverse datasets for evaluation or training (Section 3.6).

Of course, MUI is still limited by the current state of interpretability techniques, whose progress will continuously push forward a more precise model utilization evaluation.

## 2 New Metric: Model Utilization Index

### 2.1 Preliminary

Recent advancements in understanding LLMs have increasingly highlighted the importance of mechanistic interpretability (Geva et al., 2021; Wang et al., 2022; Meng et al., 2022; Conmy et al., 2023; Bricken et al., 2023), a method that aims to reverse-engineer these complex models at a more granular level. This approach goes beyond black-box testing by examining the inner workings of neural networks—investigating how individual components, such as neurons and layers, interact to produce the model's overall behavior. The goal is to uncover the causal mechanisms that drive the model's responses, offering a deeper understanding of its decision-making processes. Before introducing our proposed metric, we will briefly explore two prominent techniques in mechanistic interpretability: neuron-based and Sparse Activation Encoding (SAE) based techniques.

### 2.1.1 Neuron-Based Interpretability

Neuron-level (Dai et al., 2022; Wang et al., 2022; Gurnee et al., 2023) interpretable methods connect individual neurons in the Feed-forward network (FFN) sub-layer of LLMs to specific semantic meanings. These neurons are treated as mediator variables (Meng et al., 2022) for certain model behaviors. A causal relationship between model behaviors and neuron activations is constructed by intervention techniques, such as neuron activation patching (Vig et al., 2020; Meng et al., 2022; Chen

et al., 2024), neuron activation comparison (Wang et al., 2022), and neuron gradient comparison (Dai et al., 2022). As a large portion of neurons are shown to be connected to certain interpretable concepts (Bills et al., 2023), we use their activation footprints to measure model utilization.

Considering the large number of neurons within LLMs, before testing the causal effect, the most important step in neuron-level analyses is identifying key neurons that are activated and functioning for a certain task of interest, i.e., the actually utilized neurons in the model for performing the task. The most common pipeline for identifying key neurons is to calculate a contribution score for each neuron quantifying the contribution of the neuron to solving the task and then setting up a threshold to select the key neurons with scores over the threshold. There are various ways to calculate the contribution score, such as using gradient-based attribution (Dai et al., 2022), correlations to prediction (Wang et al., 2022), and contribution to certain tokens (Pan et al., 2023; Geva et al., 2021; 2022). In our implementation, we follow a straightforward but effective way of defining the contribution score of a neuron as its contribution to promoting the prediction of the desired output token at the position of the last token before prediction (Pan et al., 2023; Geva et al., 2021; 2022).

Specifically, omitting the layer normalization for the sake of brevity, the FFN sub-layer of layer $l$ can be considered as the function:

$$\text{FFN}^l(\mathbf{x}) = \mathbf{W}_{\text{out}}^l \, \sigma \left( \mathbf{W}_{\text{in}}^l (\mathbf{x}) \right), \tag{1}$$

where $\sigma$ is an activation function, $\mathbf{W}_{\text{in}}^l$ and $\mathbf{W}_{\text{out}}^l$ are the first/second linear layer in FFN. The contribution score of the $i$-th neuron in layer $l$ for prediction token $\hat{y}$ given input $x$ can be defined:

$$f_{\text{neuron}}(i, l, \hat{y} \mid x) = \left( \mathbf{W}_u \mathbf{W}_{\text{out}}^l \circ \sigma \left( \mathbf{W}_{\text{in}}^l \left( \mathbf{x}_{-1}^l \right) \right)^\top \right)_{i, \hat{y}}, \tag{2}$$

where $\mathbf{W}_u$ is the unembedding matrix transforming the hidden states into scores over the vocabulary, $\circ$ is an element-wise product with broadcasting, and $\mathbf{x}_{-1}^l$ denotes the input of FFN in the last token before predicting $\hat{y}$ at $l$-th layer. Consider a task sample $t = (x, y)$ from the evaluation dataset $T = \left\{ (x_1, y_1), (x_2, y_2), \ldots, (x_{|T|}, y_{|T|}) \right\}$. For a threshold $\eta$, the key neurons for task sample $t$ is:

$$N_{\text{neuron}}(t) = \left\{ (i, l) \,\middle|\, f_{\text{neuron}} \left( i, l, \hat{y}_j \mid x \bigoplus \hat{y}_{<j} \right) > \eta, \, \hat{y}_j \in y, l = 1, 2, \ldots, L, i = 1, 2, \ldots, N \right\}, \tag{3}$$

where: $\hat{y}_{<j} = (\hat{y}_1, \hat{y}_2, \ldots, \hat{y}_{j-1})$ denotes the partial response sequence before the j-th token $\hat{y}_j$, $L$ represents the total number of layers in the model, and $N$ indicates the number of neurons per layer. These key neurons represent the essential components to produce the output $y$ given $x$. In other words, they reflect the "effort" the model exerts to achieve the response. More details can be found in Appendix B.6.1.

### 2.1.2 SAE-BASED INTERPRETABILITY

Sparse-Autoencoder (SAE) is similar to neuron analysis but targets the activation patterns in each layer. By learning a sparse representation (Lieberum et al., 2024a; He et al., 2024b), the SAE effectively segregates model behaviors into distinct, interpretable feature units, thereby addressing the challenge of *polysemanticity* in neuron-level interpretation. Specifically, SAEs project the hidden states from the $l^{\text{th}}$ layer of an LLM, denoted as $\mathbf{x}^l$, into a high-dimensional feature space using a linear encoder parameterized by matrix $\mathbf{W}_e^l$. A corresponding decoder, defined by $\mathbf{W}_d^l$, reconstructs the original hidden states. To promote sparsity in the extracted features, the encoder's output undergoes post-processing with a sparsity-enforcing constraint (*e.g.,* TopK (Gao et al., 2024; He et al., 2024b), JumpReLU (Lieberum et al., 2024a)), which limits the number of non-zero values in the output, thus ensuring that only input corresponding features are retained.

$$\mathbf{f}^l = \text{SparsityConstraint} \left( \mathbf{W}_e^l \mathbf{x}^l \right), \quad \mathbf{W}_d^l \mathbf{f}^l \approx \mathbf{x}^l \tag{4}$$

The projected mono-semantic features dictionary $\mathbf{f}^l \in R^D$ contains $D$ features. We define the score of the $i^{\text{th}}$ feature in layer $l$ in relation to its contribution for prediction $\hat{y}$ for a given input x as follows:

$$f_{\text{sae}}(i, l, \hat{y} \mid x) = \text{SparsityConstraint} \left( \mathbf{W}_e^l \mathbf{x}_{-1}^l \right)_{i, \hat{y}} \tag{5}$$

For a given threshold $\eta$, the key features for completion task sample $t$ is defined as:

$$N_{\text{sae}}(t) = \left\{ (i, l) \,\middle|\, f_{\text{sae}} \left( i, l, \hat{y}_j \mid x \bigoplus \hat{y}_{<j} \right) > \eta, \, \hat{y}_j \in y, l = 1, 2, \ldots, L, i = 1, 2, \ldots, D \right\}, \tag{6}$$

## 2.2 Model Utilization Index

Now, we formally define Model Utilization Index (MUI) as follows:

$$\mathbf{MUI} = \frac{N_{\text{activated}}(T)}{N_{\text{total}}}$$

where $N_{\text{total}}$ denotes the total capabilities of the model, $N_{\text{activated}}(T)$ denotes the number of activated capabilities when the model completes the tasks $T$, e.g., samples in the test set. Now, we define the capabilities through Mechanistical interpretability techniques.

When we leverage neuron-based methods, $\mathbf{MUI}_{\text{neuron}}$ is instantiated using:

$$N_{\text{activated}}(T, \text{neurons}) = \left| \bigcup_{t=1}^{T} \{i \mid i \in N_{\text{neuron}}(t)\} \right|$$

where $N_{\text{neuron}}(t)$ is defined in Section 2.1.1. In experiments, we pick up Top $k\%$ neurons with highest activation values in each layer to setup the threshold $\eta$, to avoid the impacts of model scale.

When we leverage SAE-based methods, $\mathbf{MUI}_{\text{feature}}$ is instantiated using:

$$N_{\text{activated}}(T, \text{features}) = \left| \bigcup_{t=1}^{T} \{i \mid i \in N_{\text{sae}}(t)\} \right|$$

where $N_{\text{sae}}(t)$ is defined in Section 2.1.2. In experiments, we pick up Top $k\%$ active features in each SAE layer, to avoid the impacts of the size of pre-defined SAE features.

## 3 Experiments

### 3.1 Setup

**Dataset Selection.** To ensure reliable conclusions, we select diverse and widely used benchmarks. Following (Grattafiori et al., 2024; Ying et al., 2024b), we include 1) GSM8K (Cobbe et al., 2021) and MATH (Hendrycks et al., 2021) for math reasoning, 2) HumaEval (Chen et al., 2021) and MBPP (Austin et al., 2021) for coding, 3) ARC-Challeng (Clark et al., 2018) for science (including math) reasoning, and 4) BIG-bench Hard (BBH) (bench authors, 2023) and MMLU (Hendrycks et al., 2020) to cover general tasks. Statistical result for the selected benchmarks is shown in Table 2.

**Model Selection.** To maximize the applicability of MUI and ensure the fairness of the evaluation, we carefully select four series of widely used open-sourced LLMs: 1) Llama Series, 2) Qwen series, 3) Gemma series, and 4) OLMo series (detailed checkpoints information is shown in Appendix A.3). Note that we select merely ~7B LLMs considering the cost and those models are more probably trained well. Nevertheless, we also include some larger models in Table 5 for exploratory analysis, where the results are basically consistent with our claims. More can be found in Appx. A.4.

**Model Ranking.** In fact, to validate the effectiveness of MUI is challenging due to the lack of a universally recognized reference ranking as ground truth. For instance, in official technical reports, Google and Meta each demonstrate results surpassing their competitors. However, in independent community testing, when using the same evaluation settings, Gemma-2-9b outperforms Meta-Llama-3.1-8B on six out of seven diverse tasks (including BBH, GPQA, MUSR, etc.). Therefore, we manually curate a reference ranking considering: 1) selecting models with clearly differing fundamental capabilities, 2) model's release order, and 3) expert consensus.

### 3.2 Model Utility Law

> **Utility Law.** *Inverse Relationship Between Capability and MUI: As foundational capability increases, model utilization on a fixed dataset decreases.*

We first verify MUI by introducing a common phenomenon that is consistent with our hypothesis: the less effort required to achieve a better result, the stronger the individual's ability. We selected several fundamental LLMs, excluding those specifically optimized, such as CodeLlama, which will be

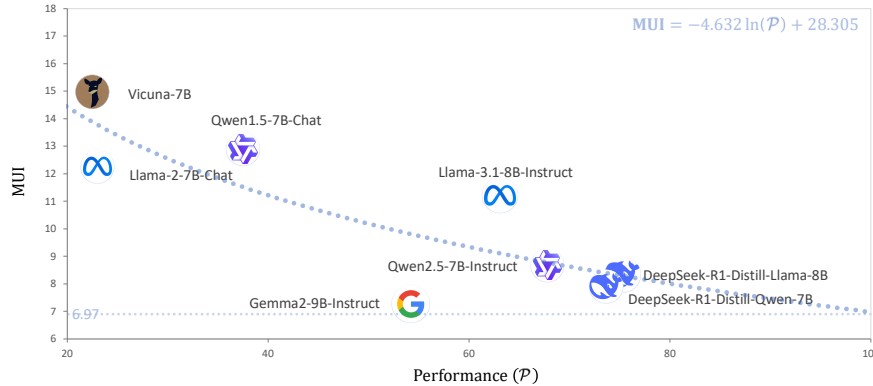

Figure 2: Overall MUI-performance relationship across six datasets. MMLU is excluded due to inference cost considerations. By fitting the curve, the minimum MUI is around 6.97% when performance reaches 100%, implying a limit sparsity optimization ratio.

analyzed later. By fitting the relationship between MUI and the corresponding performance across various datasets, we observed a universal law. Figure 2 illustrates the MUI-performance curve on aggregated datasets (curves on each separate dataset can be found in Appendix B.1). It is important to note that Figure 2 does not represent a simple weighted average across all datasets, as the number of activated neurons in the MUI formula varies with the increasing amount of data. Specifically, we observe that the MUI-performance curve exhibits an approximately negative logarithmic:

$$\mathbf{MUI} = A \ln(\mathcal{P}) + B$$

where $\mathcal{P}$ denotes performance score, and we have $A = -4.632, B = 28.305$ for overall relationship. We also observe similar trend using SAE-based MUI, which can be found in Appendix B.1.

From the above model utilization law, we can observe that: 1) From the upper left to the lower right, the ranking of model capabilities generally aligns with expectations. In particular, for models with similar performance, such as Vicuna and Llama2, MUI distinctly differentiates their ability levels. A more detailed analysis is provided in Section 3.5 (Corollary 3); 2) There are two special points on the logarithmic curve. First, as performance approaches zero, MUI tends toward infinity, implying that evaluation loses its meaning; second, when the performance score reaches 100%, MUI approaches

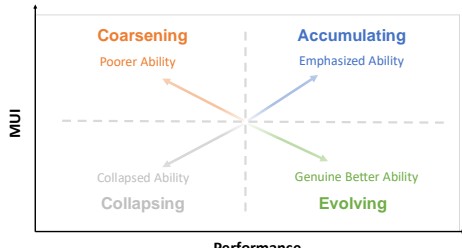

Figure 3: Four optimization directions.

approximately 6.97%. This may indicate a limit compression ratio — if a lower MUI is preferable, then this value represents the minimum expected model utilization, providing training guidance. With the development of dataset and interpretability techniques, this value could be more precise.

### 3.3 MUI FOR MODEL TRAINING

> **Corollary 1.** *Training Diagnostics: MUI-performance curve implies four types of optimization processes during training: evolving, accumulating, coarsening, and collapsing.*

As mentioned above, in the MUI-performance curve, the progression from the upper left to the lower right represents an improvement in the model's fundamental capability: lower MUI coupled with higher performance. Inspired by this, we further explore the other directions to analyze the training process of a model for training diagnostics. As shown in Figure 3, we first define four types of optimization directions: 1) **Evolving**: The model demonstrates improved performance with a smaller MUI, indicating genuinely enhanced capability. 2) **Accumulating**: The model shows better performance but with an increased MUI, suggesting the improvements on a particular ability, which usually happens in the early stage of learning. 3) **Coarsening**: The model exhibits poorer

performance with an increased MUI, which indicates some reduced abilities seemingly as a side-effect of accumulating. 4) **Collapsing**: The model displays poorer performance with a smaller MUI, suggesting a comprehensive breakdown in model functionality.

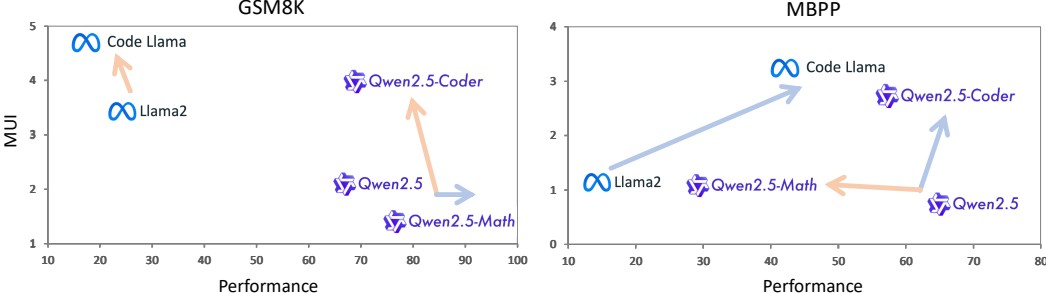

Figure 4: Optimization directions in MUI-Performance dimensions, studied on Llama / Qwen series. We compare the math / code versions with the base models, to see the changes using in-domain testing and out-of-domain testing (e.g., for code version, MBPP/GSM8K is in/out-domain).

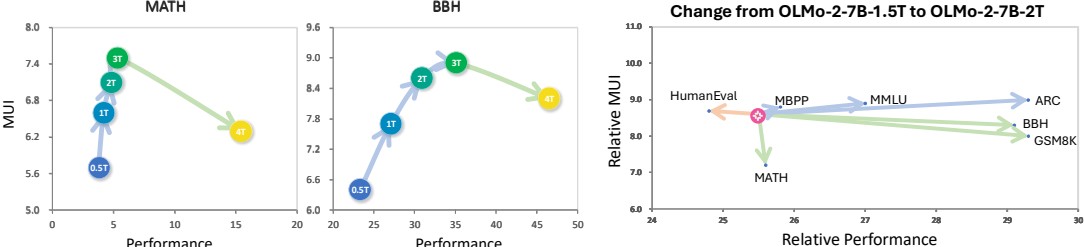

Figure 5: MUI-Performance curves for OLMo-2-7B pre-trained using 0.5T, 1T, 2T, 3T, 4T tokens.

Figure 6: Relative MUI-Performance curve for OLMo-2-7B-1.5T&2T data.

We now empirically demonstrate these four directions based on existing LLMs, followed by investigating the pre-training progress of the OLMo series thanks to its open-source details. First, for the evolving direction, we have demonstrated it in Section 3.2. Similar trend can also be found in Appendix Figure 11. Second, for the coarsening and accumulating directions, we assume that these two typically occur during the early stage of learning specific abilities. We thus choose math reasoning and coding as two example abilities, observing the Llama series: Llama-2-7B-Chat and CodeLlama-7B-Instruct, where the latter is tuned on coding data based on the former LLM, and the Qwen series: Qwen2.5-7B-Instruct and its coding/math optimized versions, Qwen2.5-Coder-7B-Instruct and Qwen2.5-Math-7B-Instruct.

Figure 4 first shows the **post-training analysis**, we can see that the specialized versions move in the accumulating direction compared with the base model if testing on the datasets for targeted abilities, e.g., CodeLlama vs Llama2 on MBPP, Qwen2.5-Coder vs Qwen2.5 on MBPP, and Qwen2.5-Math vs Qwen2.5 on GSM8k, while the specialized versions move in the coarsening direction compared with the base model if testing on the out-of-domain (OOD) benchmark, e.g., CodeLlama vs Llama2 on GSM8k, Qwen2.5-Coder vs Qwen2.5 on GSM8k, and Qwen2.5-Math vs Qwen2.5 on MBPP. We attribute the main reason is when the performance improvement on a particular task has not yet reached the level of fundamental capability enhancement — primarily due to an overall distributional shift towards that ability — it corresponds to the accumulating direction (with an increase in MUI and performance). Simultaneously, this shift results in decreased performance on other tasks, characterizing the coarsening direction (with an increase in MUI and a decrease in performance). Full results on the selected tasks shown in Appendix Table 5 and Table 9. This observation in turn suggests that using MUI as an indicator on one fixed dataset can relate to changes in other capabilities, a.k.a., generalizable evaluation beyond current benchmarks. Third, for the collapsing direction, we only observe it when data contamination occurs, detailed in Section 3.4.

For **pre-training**, we select a series of OLMo-2-7B checkpoints, ranging from 0.5T to 4T training data. Figure 5 shows similar curves when testing using different datasets, e.g., MATH and BBH, where full results are in Appendix Figure 12 and Table 8. We can see that the model initially exhibits "accumulating", progressing until the final 200k steps (3T to 4T), whereas evolving trend becomes uniformly evident. By investigating intermediate steps during accumulating (Figure 6), we observe

it accompanied by both coarsening and evolving trends on different tasks, which indicates different capabilities being enhanced separately, eventually leading to a comprehensive enhancement — the evolving direction. By closely monitoring changes in MUI on limited datasets, we can identify overall trends in model capabilities improvement and make timely, targeted adjustments. We thus summarize it as Corollary 1 beneficial for training guidance.

### 3.4 MUI FOR DATA CONTAMINATION

> **Corollary 2.** *Data Contamination Analysis: When a model achieves inflated performance through data contamination, it does not reduce model utilization; instead, utilization increases as other capabilities are compromised.*

In the previous section, we showed that optimizing a model for a single capability drives its MUI-performance curve along the coarsening and accumulating directions. Here, we examine how the curve changes under data contamination. To simulate this issue, we fine-tuned three models: Llama-2-7B-Chat, Llama-3.1-8B-Instruct, and Qwen-2.5-7B-Instruct, using the test samples of GSM8K and MATH following (Ying et al., 2024a), detailed hyper-parameters are listed in Appendix A.7. The results are shown in Figure 7, and full results with similar trends in Table 10.

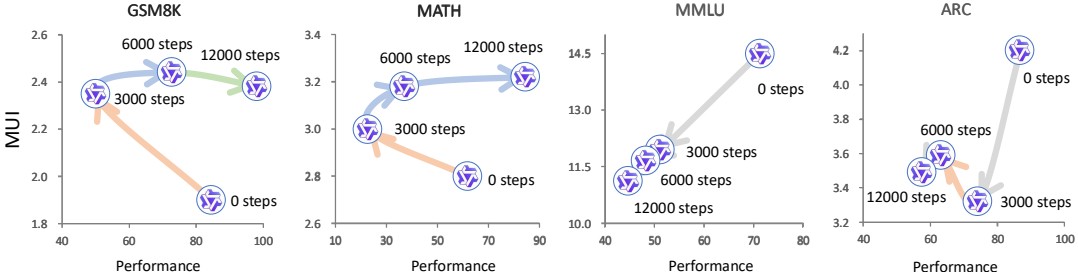

Figure 7: Simulation of data contamination by tuning Qwen2.5-7B-Instruct on test samples from GSM8K and MATH, where ARC and MMLU can be regarded as OOD testing.

We observe that 1) the models show both higher MUI and performance on the contaminated datasets. 2) Specifically, they first move in the coarsening direction — both performance and MUI decrease. As training steps increase, they gradually shift toward the accumulating direction and, on GSM8K, even show an evolving tendency. This trajectory mirrors the pattern induced by targeted capability optimization in the section 3.3. The resemblance is unsurprising: when leaked data come from the same domain, they effectively boost the model's in-domain competence; with enough leaked samples, the effect becomes indistinguishable from specific ability enhancement. 3) By contrast, on OOD testing (i.e., ARC and MMLU), the models move along the collapsing direction. This reveals the key difference between data contamination and capability-specific optimization: overfitting to limited leaked samples raises MUI by encroaching on neurons responsible for OOD abilities, leading to simultaneous declines in both performance and MUI. That's why we name this direction as collapsing. Therefore, data contamination and targeted capability optimization exhibit the same underlying dynamics; thus, Corollary 2 is not limited to contamination scenarios.

**Takeaway:** In summary, the ideal optimization trajectory during pre-training or post-training is the evolving direction. In the early stage of learning, an accumulating trend signals insufficient training and warrants continued training; if it is accompanied by some coarsening, the data mix should be promptly rebalanced based on the fine-grained categories in the evaluation set. By contrast, the emergence of a collapsing direction calls for an immediate halt to training and a thorough inspection of the training and test sets for severe overfitting.

### 3.5 MUI FOR MODEL COMPARISON

> **Corollary 3.** *Model Comparison & Ranking: On a fixed dataset, stronger foundational capabilities should be reflected by both higher performance and lower model utilization.*

In the previous sections, we present the common phenomenon revealed by the MUI–performance curve. In this section, we quantitatively demonstrate the effectiveness and stability of MUI from

| Model | GSM8K | MATH | ARC$_c$ | HumanEval | MBPP | BBH | Ref./Avg Correlation |
|---|---|---|---|---|---|---|---|
| Vicuna-7B | 11.9 / 5.1 | 4.0 / 1.7 | 54.0 / 19.6 | 13.4 / 12.2 | 23.6 / 17.6 | 28.6 / 9.9 | 9 |
| Llama-2-7B-Chat | 25.8 / 13.2 | 5.4 / 2.4 | 60.0 / 25.8 | 16.4 / 17.3 | 24.8 / 21.0 | 26.2 / 9.0 | 8 |
| Qwen1.5-7B-Chat | 58.7 / 33.9 | 17.9 / 9.4 | 76.2 / 30.6 | 39.0 / 41.1 | 39.8 / 33.6 | 41.3 / 14.2 | 7 |
| Llama-3-8B-Instruct | 79.5 / 43.8 | 26.6 / 11.7 | 82.1 / 34.4 | 59.8 / 54.6 | 56.5 / 41.0 | 64.0 / 23.9 | 6 |
| Llama-3.1-8B-Instruct | 86.5 / 52.6 | 48.1 / 24.7 | 81.9 / 42.0 | 62.2 / 56.8 | 57.9 / 39.0 | 65.4 / 23.6 | 5 |
| Gemma-2-9B-Instruct | 82.3 / 55.5 | 33.5 / 18.7 | 89.5 / 47.2 | 63.4 / 75.8 | 61.2 / 58.4 | 57.5 / 26.3 | 4 |
| Qwen2.5-7B-Instruct | 84.5 / 61.3 | 61.9 / 37.0 | 86.8 / 42.4 | 71.3 / 85.2 | 62.1 / 62.1 | 66.0 / 28.7 | 3 |
| DeepSeek-Llama3.1-8B | 71.6 / 40.7 | 74.1 / 39.1 | 82.6 / 43.5 | 68.9 / 72.6 | 60.3 / 49.2 | 76.8 / 33.4 | 2 |
| DeepSeek-Qwen2.5-7B | 82.6 / 51.2 | 80.1 / 42.2 | 81.1 / 44.0 | 71.9 / 85.9 | 62.5 / 57.1 | 66.5 / 30.4 | 1 |
| Spearman | 68.3 / 68.3 | 98.3 / 98.3 | 66.7 / 90.0 | 98.3 / 95.0 | 95.0 / 85.0 | 91.7 / 95.0 | $86.4_{\Delta 1.8}$ / $88.6_{\Delta 1.0}$ |
| Kendall | 55.6 / 61.1 | 94.4 / 94.4 | 50.0 / 83.3 | 83.3 / 94.4 | 88.9 / 72.2 | 77.8 / 83.3 | $76.9_{\Delta 3.2}$ / $80.6_{\Delta 1.2}$ |

Table 1: Accuracy/PUR score (%) across six datasets. We exclude MMLU due to the inference cost. Ref is short for reference rank. All Spearman or Kendall coefficients are with $< 0.02$ p-value.

the aspect of model comparison. However, it is difficult to find out a universally accepted baseline for model ranking; for example, our observations show that even the Arena score is strongly biased toward response style and format, or highly correlated with mathematical reasoning tasks — exhibiting up to a 90% Pearson correlation with the GSM8K leaderboard regarding our selected models. Therefore, we manually order nine base models as reference. Next, we design a simple composite metric that integrates MUI with performance and use it to reorder the same nine models. We then compute the correlation and variance between the two rankings: a higher correlation indicates a more reasonable ordering, while a smaller variance implies greater metric stability. Note that this composite metric is introduced solely for experimental convenience; devising a more principled aggregate metric is left to future work. Specifically, the combined metric is defined as the ratio of performance to MUI (PUR):

$$\text{PUR} = \frac{\mathcal{P}}{\text{MUI}^{\alpha}} \tag{7}$$

where $\alpha$ is a hyperparameter to punish MUI for balanced scale. We set it to $0.5$ in experiments.

Table 1 shows the overall performance and PUR values on six datasets. We establish two independent rankings of the nine LLMs. The first ranking is derived solely from raw performance and serves as the baseline that reflects conventional evaluation practice. The second ranking is obtained from the PUR scores, which jointly integrate MUI and performance. We then compute the correlation between each ranking and the manually curated reference. To quantify the agreement, we adopt Spearman's correlation, which measures the strength of any monotonic relationship between two ranked lists, and Kendall's coefficient, which evaluates the consistency of all pairwise orderings and is generally more robust to ties and outliers. Our findings are as follows. 1) Higher correlation. The PUR-based ranking aligns more closely with the reference than the performance-based ranking, achieving average correlations of 88.6% versus 86.4% under Spearman, and 80.6% versus 76.9% under Kendall, thereby providing quantitative evidence of its effectiveness. 2) Lower variance. The PUR-based ranking exhibits smaller variance across datasets: 1.0 versus 1.8 (Spearman) and 1.2 versus 3.2 (Kendall), indicating that PUR delivers more stable ordering. 3) Qualitative improvements. Inspection of several ambiguous or previously mis-ranked cases (see Figure 2) shows how PUR resolves inconsistencies. For instance, although Vicuna and Llama-2 achieve nearly identical performance, MUI reveals Llama-2's substantially greater underlying capability, allowing PUR to distinguish them; a similar clarification arises between Llama-3.1 and Qwen-2.5. Likewise, while Gemma-2 is generally regarded as stronger than Llama-3.1, a performance-only ranking suggests the opposite, whereas the PUR-based ranking corrects this by incorporating MUI.

### 3.6 MUI FOR DATA DIVERSITY

> **Corollary 4.** *Positive Correlation Between Data Diversity and MUI: As data diversity increases, model utilization exhibits an upward trend.*

In the preceding sections we focused on the role of MUI in model evaluation; this section turns to data evaluation. Because MUI fundamentally measures the extent to which a test sample activates a model's latent abilities, fixing the model allows us to invert the perspective: MUI can reveal which samples trigger different capabilities, serving as a model-specific indicator of data diversity.

There is no universally accepted definition of data diversity. Most existing work pursues diversity by balancing samples across capabilities or domains. We therefore experimentally examine the relationship between MUI and them. Note that a higher MUI indicates that a wider range of abilities

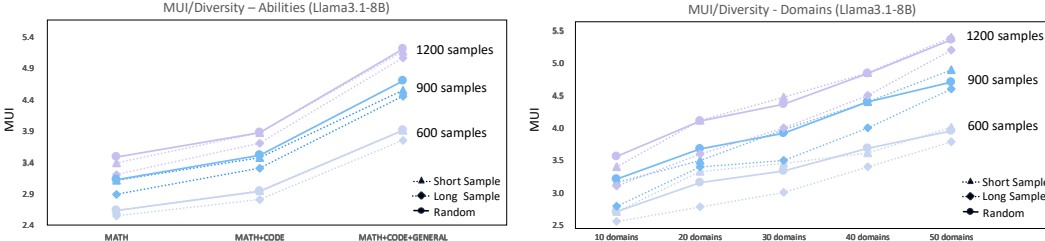

Figure 8: MUI across different data diversity dimensions: abilities and domains. Long or short samples are differentiated by 50% maximum length. The Mann-Whitney U test, at a 95% confidence level, the statistical significance analysis reveal that there are no statistically significant differences in proportions between short and long texts across the various dataset sizes.

has been activated, i.e., more diverse data. All experiments use Llama-3.1-7B-Instruct as the reference model and neuron-based MUI. Comparable findings for additional models, an analysis based on SAE interpretation, and experiments on more dimensions are provided in Appendix B. For capability diversity, we following earlier experiments and consider three capabilities — mathematics, coding, and general abilities. From datasets covering different capability combinations, we randomly draw a fixed number of samples and compute their MUI, the results are shown in Figure 8. We can see that: 1) Overall, MUI shows a positive correlation with capability diversity. 2) Specifically, for similar capability combinations (from math to math+code), MUI grows slowly, whereas it rises rapidly when the combined capabilities are more dissimilar. 3) Roughly 600 samples spanning all three capabilities yield a MUI comparable to 1200 samples drawn from a single capability set, suggesting that MUI-guided sampling can substantially boost the efficiency of diversity selection. For domain diversity, we leverage existing labels from MMLU, BBH, MATH, and ARC. The results show similar trends. MUI is positively correlated with domain diversity. However, when we further stratify samples by length or difficulty (Appendix B.4), no significant trend emerges, implying that these factors are largely independent of the breadth of abilities a model can express.

## 3.7 LIMITATION

By leveraging the two mechanistic interpretability techniques (i.e., neuron-based and SAE-based), we uncovered the Utility Law and its four corollaries, thereby demonstrating the viability of using MUI for both model- and data-level evaluation. Nevertheless, such a new metric may be limited in several aspects such as: 1) the robustness of our method on answer correctness, and 2) different neuron-probing methods and selection standards. Due to space limitation, ablation studies are in Appendix B.5 and Appendix B.6, which shows no significant difference from our findings. Besides, interpretability remains a rapidly evolving field, and current technical constraints can make such extensions either difficult to implement or prone to metric instability. For example, when we switch to an SAE-based MUI, the scarcity of publicly available SAE models, the high cost of training new SAEs, and the need for layer-specific SAEs across different models restrict us to a very limited experimental set and preclude fair cross-model comparison. Likewise, replacing our present neuron-probing method still leaves the Model Utilization Law largely intact (We obtain a preliminary average Pearson correlation coefficient of 0.78), but a larger number of outliers emerge. These suggest that further investigation is needed on both fields of evaluation and interpretability.

## 4 CONCLUSION

We introduced MUI — a mechanism interpretable metric that gauges how efficiently an LLM uses its capacity. Combined with performance, MUI offers an effective, stable, and generalizable analysis and evaluation for both model and data. Across various base models and benchmarks, we show a stable inverse, near-log MUI-performance curve, formalized as the Utility Law as well as four corollaries. Based on them, we discover four optimization directions during training — evolving, accumulating, coarsening, and collapsing — which clarify capability gains, specialization trade-offs, and data-contamination impacts. For model comparison, MUI produces model rankings that align closely with expert judgment while remaining variance-robust. For data diversity, we also show a positive correlation between MUI and various dimensions like domains and abilities, suggesting our metric as a model-specific comprehensive measurement. In the future, we are interested in generalizable evaluation for training guidance.

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

## A  EXPERIMENT DETAILS

### A.1  DATASET STATISTICAL RESULT

Following (Grattafiori et al., 2024), we focus on three key abilities: Mathematical and Reasoning, Coding, and General Capability. For each ability, we select several publicly available datasets to explore the Utility Law. Table 2 provides a detailed summary of the statistical characteristics of the selected datasets.

| Model | GSM8K (Math & Reasoning) | MATH (Math & Reasoning) | ARC$_c$ (Math & Reasoning) | HumanEval (Code) | MBPP (Code) | BBH (General) | MMLU (General) | Totally |
|---|---|---|---|---|---|---|---|---|
| # Testing Samples | 1,319 | 5,000 | 1,172 | 164 | 500 | 6,511 | 14,042 | 28,708 |

Table 2: The statistical detail of the selected benchmarks.

### A.2  BASELINE MODEL SELECTION

We carefully select four series of widely used open-sourced LLMs: 1) Llama Series including Vicuna-7B-v1.3 (Peng et al., 2023), Llama-2-7B-Chat (Touvron et al., 2023), Llama-3.1-8B, Llama-3.1-8B-Instruct (Grattafiori et al., 2024), CodeLlama-7B-Instruct (Rozière et al., 2024) and DeepSeek-R1-Distill-Llama-8B (DeepSeek-AI, 2025), 2) Qwen series including Qwen1.5-7B-Chat (Bai et al., 2023), Qwen2.5-7B-Instruct (Team, 2024b), Qwen2.5-Coder-7B-Instruct (Hui et al., 2024), Qwen2.5-Math-7B-Instruct (Yang et al., 2024) and DeepSeek-R1-Distill-Qwen-7B (DeepSeek-AI, 2025), 3) Gemma series including Gemma-2-9B and Gemma-2-9B-Instruct (Team, 2024a), 4) OLMo series including several checkpoints from OLMo-2-7B (OLMo et al., 2024) (detailed checkpoints information is shown in Appendix A.3).

### A.3  OLMO SERIES MODEL SELECTION

For OLMo (OLMo et al., 2024) series model, we include eight checkpoints detailed in Table 3.

| Custom Checkpoint Name | Original Checkpoint Name | Training Steps | Training Tokens | Traning Stage |
|---|---|---|---|---|
| OLMo-2-7B-0.5T | stage1-step122000-tokens512B | 122,000 | 512B | 1 |
| OLMo-2-7B-1T | stage1-step244000-tokens1024B | 244,000 | 1,024B | 1 |
| OLMo-2-7B-1.5T | stage1-step366000-tokens1536B | 366,000 | 1,536B | 1 |
| OLMo-2-7B-2T | stage1-step488000-tokens2047B | 488,000 | 2,047B | 1 |
| OLMo-2-7B-2.5T | stage1-step610000-tokens2559B | 610,000 | 2,559B | 1 |
| OLMo-2-7B-3T | stage1-step732000-tokens3071B | 732,000 | 3,071B | 1 |
| OLMo-2-7B-3.5T | stage1-step855000-tokens3587B | 855,000 | 3,587B | 1 |
| OLMo-2-7B-4T | OLMo-2-1124-7B | 928,646 | 3,896B | 1 |

Table 3: Summary of the checkpoints of model OLMo-2-1124-7B used in the study. "Custom Checkpoint Name" represents simplified names defined in this paper for clarity.

### A.4  MECHANISTIC INTERPRETABILITY TECHNIQUES

For neuron analysis, we primarily follow one of the most commonly used methods, as described in Section 2.1.1, due to time and cost considerations. However, we emphasize that our approach is not limited to a fixed neuron analysis method. Instead, we aim to explore MUI using multiple techniques, to ensure a comprehensive and robust analysis. In our ablation study in Section B.6.4, we deploy other neuron analysis methods to further investigate and validate our findings. When using the method defined in Section 2.1.1, the response $y_i$ in Equation 3, are generated under specific conditions depending on the benchmark. For BBH, 3-shot examples from the original benchmark are used. For all other benchmarks, responses are generated in a zero-shot manner for instruction-tuned models, while a human-crafted one-shot setting is used for all the base models. Details of the model generate configuration and the few-shot examples are provided in Appendix A.7. The $\eta$ in Equation 3, is set to the top 1‰ of key neurons and selected at the layer level (corresponding to the

top 1‰ of $N$). This threshold function is detailed as follows:

$$N_{\text{neuron}}(t) = \left\{ (i,l) \;\middle|\; f_{\text{neuron}}(i,l,\hat{y}_j \mid x + \hat{y}_{<j}) \geq V_l^{top1‰}, \hat{y}_j \in y, l \in \{1,2,\dots,L\}, i \in \{1,2,\dots,N\} \right\}, \quad (8)$$

where :$V_l = [f_{\text{neuron}}(i,l,\hat{y}_j \mid x + \hat{y}_{<j}) \mid \hat{y}_j \in y, i \in \{1,2,\dots,N\}]$, with $L$ representing the number of layers and $N$ the number of neurons in the tested model.

Regarding using SAE to conduct analysis: we utilized all publicly available LLMs with SAEs for our analysis, including Llama3.1-8B with Llama Scope SAE (He et al., 2024a) and Gemma-2-9B & Gemma-2-9B-Instruct with SAE from Gemma Scope (Lieberum et al., 2024b). The response generation process for $y_i$ in Equation 6 follows the same procedure as described in the neuron analysis technique. To ensure a convenient and as fair as possible comparison of model utilization across different architectures, we selected the Residual SAE with the width of 128K for each possible layer. Considering that the SAEs trained by Llama Scope use the Top50-ReLU activation function, while those from Gemma Scope adopt JumpReLU, we adjusted the selection criteria for the Gemma SAEs. Specifically, for each layer in Gemma, we selected SAEs with an $L_0$ close to 50 to align with the Llama Scope configuration. Detailed parameter selection is shown in Appendix A.6. The $\eta$ in Equation 6, is set to the top 50 of features and selected at the layer level. This threshold function is detailed as follows:

$$N_{\text{sae}}(t) = \left\{ (i,l) \;\middle|\; f_{\text{sae}}(i,l,\hat{y}_j \mid x + \hat{y}_{<j}) \geq V_l^{top50}, \hat{y}_j \in y, l \in \{1,2,\dots,L\}, i \in \{1,2,\dots,D\} \right\}, \quad (9)$$

where :$V_l = [f_{\text{sae}}(i,l,\hat{y}_j \mid x_i + \hat{y}_{<j}) \mid \hat{y}_j \in y, i \in \{1,2,\dots,D\}]$, with $L$ representing the number of layers and $D$ the number of featuresin Equation 4.

## A.5 Implementation for Threshold Function

Generally, there are several different threshold functions implemented for $\eta$, two of which have been introduced in Appendix A.4. Here, we uniformly represent these methods within the context of neuron and SAE interpretation:

- **Top-k:** that we select neurons that have the top $k$ values for each layer, as defined by:

$$N_{\text{neuron / sae}}(t) = \left\{ (i,l) \;\middle|\; f_{\text{neuron / sae}}(i,l,\hat{y}_j \mid x + \hat{y}_{<j}) \geq V_l^{topk}, \hat{y}_j \in y, l \in \{1,2,\dots,L\}, i \in \{1,2,\dots,N/D\} \right\}, \quad (10)$$

  where, $V_l = [f_{\text{neuron / sae}}(i,l,\hat{y}_j \mid x_i + \hat{y}_{<j}) \mid \hat{y}_j \in y, i \in \{1,2,\dots,N/D\}]$

- **Global Top-k:** is a similar approach but selecting top values across all layers:

$$N_{\text{neuron / sae}}(t) = \left\{ (i,l) \;\middle|\; f_{\text{neuron / sae}}(i,l,\hat{y}_j \mid x + \hat{y}_{<j}) \geq V^{topk}, \hat{y}_j \in y, l \in \{1,2,\dots,L\}, i \in \{1,2,\dots,N/D\} \right\}, \quad (11)$$

  where $V = [f_{\text{neuron / sae}}(i,l,\hat{y}_j \mid x_i + \hat{y}_{<j}) \mid \hat{y}_j \in y, l \in \{1,2,\dots,L\}, i \in \{1,2,\dots,N/D\}]$

- **Top-%k:** similar to the top-k method, this approach uses a fixed percentage, $k\%$, multiplied by $N/D$ to determine the threshold. This modification can alleviate discrepancies due to different model architectures when calculating the MUI. Specifically, when calculating the MUI, if the denominator $N_{\textbf{total}}$ is defined as $N_{\textbf{total}} = N/D \times L$, setting the threshold as a proportion of $N/D$ helps mitigate issues caused by varying model structures. This ensures a more equitable comparison across various models.

- **Top-score:** as dai2022knowledge, using a fraction $k$ of the maximum attribution score for each layer:

$$N_{\text{neuron / sae}}(t) = \left\{ (i,l) \;\middle|\; f_{\text{neuron / sae}}(i,l,\hat{y}_j \mid x + \hat{y}_{<j}) \geq kV_l^{max}, \hat{y}_j \in y, l \in \{1,2,\dots,L\}, i \in \{1,2,\dots,N/D\} \right\}, \quad (12)$$

  where $V_l^{max}$ is the maximum value in $V_l$.

Here to notice, $V_l$ and $V$ are token-level calculation, if using a sponse-level score (Equation 23) the $V_l$ will be as $V_l = [f_{\text{neuron / sae \_sum}}(i,l,y \mid x) \mid i \in \{1,2,\dots,N/D\}]$ correspondingly.

| Models | Llama-3.1-8B-Base | Gemma-2-9B-Base |
|---|---|---|
| Trained BY | Llama Scope | Gemma Scope |
| SAE Position (Layer) | Every Layer | Every Layer |
| SAE Position (Site) | Residual | Residual |
| SAE Width (# Features) | 128K | 128K |
| Activation Function | TopK-ReLU | JumpReLU |

Table 4: An overview of selected SAEs on Large Language Models.

### A.6 SPARSE-AUTOENCODER (SAE) SELECTION

Table A.6 shows the detailed parameter information of the selected SAEs.

### A.7 MODEL PARAMETER SETTING

- Response Generation: across all involved models is conducted with a fixed temperature of 0.0 (*do_sample=False*) and a maximum token length of 1024 (8192 for DeepSeek-Llama3.1-8B and DeepSeek-Qwen2.5-7B). The generation conditions vary depending on the benchmark. For BBH, 3-shot examples from the original benchmark are used. For all other benchmarks, responses are generated in a zero-shot manner for instruction-tuned models, while a human-crafted one-shot setting is applied for base models. The human-crafted few-shot example is shown in Appendx C.

- SAE Analysis: To ensure consistency and fairness in our evaluations, we truncate inputs to a maximum token length of 2048, as dictated by Llama Scope's encoder limitations. Consequently, tasks with few-shot examples from the BBH dataset, which require substantial response truncation, are excluded from our analysis.

- Full Parameter Fine-tuning: Following work (Ying et al., 2024a), we train the model on selected code-related tasks (HumanEval and MBPP) and math-related tasks (GSM8K and MATH). During full parameter fine-tuning (STF), test samples are concatenated with answers. This fine-tuning process involves adjusting several parameters: the learning rate, which varies from $7 \times 10^{-5}$ to $2 \times 10^{-9}$; the number of epochs, ranging from 2 to 4; and the batch size, set between 4 and 128. The optimal performance for each model configuration are identified. The experiment is conducted on 8 NVIDIA H20 GPUs.

### A.8 COMPUTATIONAL RESOURCES

All experiments were conducted on 8 NVIDIA H20 GPUs, each with 96GB of memory. Model inference required approximately 80 GPU-hours per model with around 7 billion parameters, whereas the DeepSeek distilled model required approximately 1920 GPU-hours. The MUI calculation using neuron analysis consumed approximately 96 GPU-hours per model of around 7 billion parameters, while SAE analysis required about 160 GPU-hours. Full parameter fine-tuning consumed approximately 144 GPU-hours.

## B MORE EXPERIMENT RESULTS

### B.1 UTILITY LAW: THE RELATIONSHIP BETWEEN MUI AND PERFORMANCE

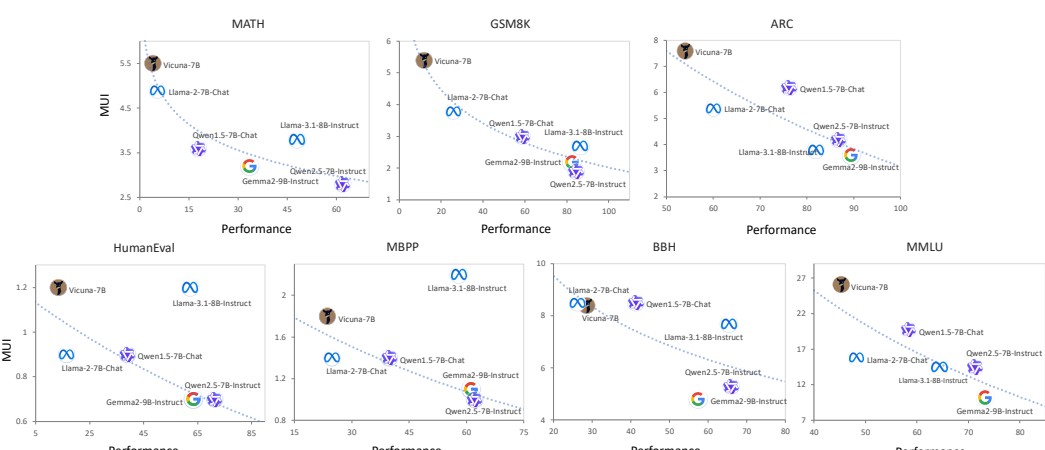

Figure 9: Performance (accuracy %) and Model Utilization Index (MUI) (%), as determined by neuron analysis (refer to Section 2.1.1), of the selected six models across the selected seven tasks. Results indicate that models with stronger performance activate fewer features for the corresponding tasks. The dashed line represents the trend line fitted using a logarithmic function.

| Model | GSM8K (Math & Reasoning) | MATH (Math & Reasoning) | $ARC_c$ (Math & Reasoning) | HumanEval (Code) | MBPP (Code) | BBH (General) | MMLU (General) |
|---|---|---|---|---|---|---|---|
| Vicuna-7B | 11.9 / 5.4 | 4.0 / 5.5 | 54.0 / 7.6 | 13.4 / 1.2 | 23.6 / 1.8 | 28.6 / 8.4 | 45.3 / 26.1 |
| Llama-2-7B-Chat | 25.8 / 3.8 | 5.4 / 4.9 | 60.0 / 5.4 | 16.4 / 0.9 | 24.8 / 1.4 | 26.2 / 8.5 | 48.2 / 15.8 |
| CodeLlama-7B-Instruct | 23.0 / 4.5 | 6.0 / 7.0 | 49.5 / 8.7 | 34.1 / 1.8 | 44.7 / 2.9 | 21.8 / 8.3 | 41.4 / 24.5 |
| Qwen1.5-7B-Chat | 58.7 / 3.0 | 17.9 / 3.6 | 76.2 / 6.2 | 39.0 / 0.9 | 39.8 / 1.4 | 41.3 / 8.5 | 58.3 / 19.8 |
| Llama-3.1-8B-Instruct | 86.5 / 2.7 | 48.1 / 3.8 | 81.9 / 3.8 | 62.2 / 1.2 | 57.9 / 2.2 | 65.4 / 7.7 | 64.4 / 14.5 |
| Gemma2-9B-Instruct | 82.3 / 2.2 | 33.5 / 3.2 | 89.5 / 3.6 | 63.4 / 0.7 | 61.2 / 1.1 | 57.5 / 4.8 | 73.3 / 10.2 |
| Qwen2.5-7B-Instruct | 84.5 / 1.9 | 61.9 / 2.8 | 86.8 / 4.2 | 71.3 / 0.7 | 62.1 / 1.0 | 66.0 / 5.3 | 71.3 / 14.5 |
| Qwen2.5-Coder-7B-Instruct | 79.6 / 3.7 | 49.5 / 6.2 | 83.8 / 9.4 | 75.0 / 1.5 | 65.9 / 2.4 | 57.8 / 11.0 | 65.6 / 30.3 |
| Qwen2.5-Math-7B-Instruct | 92.1 / 1.9 | 76.7 / 3.3 | 66.3 / 3.8 | 53.0 / 0.7 | 47.9 / 1.1 | 29.1 / 2.9 | 51.2 / 8.5 |
| DeepSeek-Llama3.1-8B | 71.6 / 3.1 | 74.1 / 3.6 | 82.6 / 3.6 | 68.9 / 0.9 | 60.3 / 1.5 | 76.8 / 5.3 | - |
| DeepSeek-Qwen2.5-7B | 82.6 / 2.6 | 80.1 / 3.6 | 81.1 / 3.4 | 71.9 / 0.7 | 62.5 / 1.2 | 66.5 / 4.8 | - |
| Vicuna-13B | 22.4 / 4.1 | 5.0 / 4.7 | 61.9 / 7.5 | 17.1 / 1.1 | 27.6 / 1.7 | 42.3 / 6.8 | 50.9 / 24.5 |
| Llama-2-13B-Chat | 36.2 / 3.5 | 7.4 / 4.1 | 66.8 / 5.0 | 18.9 / 0.9 | 29.7 / 1.4 | 26.1 / 7.9 | 53.6 / 14.0 |
| Qwen1.5-14B-Chat | 70.7 / 4.5 | 25.6 / 6.0 | 86.3 / 8.8 | 50.6 / 1.5 | 49.9 / 2.4 | 50.3 / 11.7 | 65.4 / 30.3 |
| Qwen2.5-14B-Instruct | 84.7 / 4.9 | 60.9 / 6.7 | 91.5 / 8.5 | 68.3 / 1.4 | 63.3 / 2.3 | 64.0 / 11.1 | 77.2 / 26.9 |
| Qwen2.5-Coder-14B-Instruct | 86.3 / 4.1 | 55.7 / 6.6 | 88.4 / 8.9 | 81.1 / 1.6 | 67.3 / 2.6 | 71.7 / 10.9 | 71.0 / 28.3 |
| Vicuna-33B | 35.7 / 3.1 | 8.1 / 4.3 | 67.1 / 9.4 | 17.7 / 1.1 | 25.1 / 1.6 | 46.0 / 6.6 | 51.5 / 34.8 |
| Qwen2.5-32B-Instruct | 86.7 / 2.9 | 67.3 / 2.1 | 92.2 / 5.6 | 72.0 / 1.0 | 69.9 / 1.4 | 76.4 / 3.7 | 78.5 / 19.1 |

Table 5: Performance (accuracy %) and Model Utilization Index (MUI) (%), as determined by neuron analysis. More detailed experiment settings can be found in Appendix A.4.

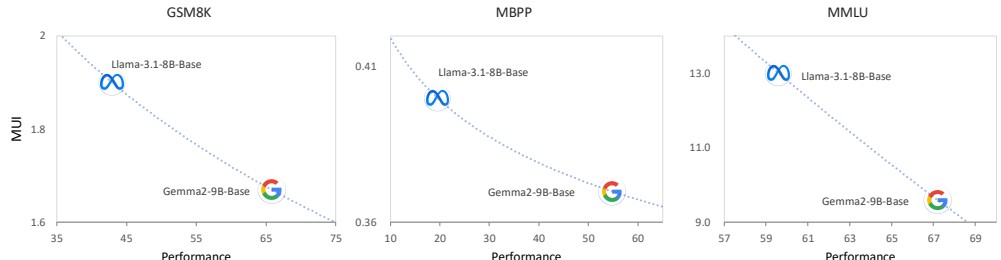

Figure 10: Relationship between performance (accuracy) and SAE-based MUI. The dashed line represents the trend line fitted using a logarithmic function. More results can be found in Table 6.

| Model | GSM8K (Math & Reasoning) | MATH (Math & Reasoning) | ARC$_c$ (Math & Reasoning) | HumanEval (Code) | MBPP (Code) | BBH (General) | MMLU (General) |
|---|---|---|---|---|---|---|---|
| Llama-3.1-8B | 42.9 / 1.9 | 15.1 / 2.0 | 77.1 / 3.2 | 19.5 / 0.40 | 26.1 / 0.7 | 50.1 / 3.9 | 59.6 / 13.0 |
| Gemma2-9B | 65.8 / 1.7 | 20.8 / 2.4 | 84.6 / 3.2 | 54.8 / 0.37 | 57.7 / 0.5 | 67.4 / 2.2 | 67.2 / 9.6 |

Table 6: Performance (accuracy %) under one-shot inference setting / SAE-based Model Utilization Index (MUI) (%) of the selected model. Detailed experiment settings can be found in Appendix A.4.

| Model | GSM8K (Math & Reasoning) | MATH (Math & Reasoning) | ARC$_c$ (Math & Reasoning) | HumanEval (Code) | MBPP (Code) | BBH (General) | MMLU (General) |
|---|---|---|---|---|---|---|---|
| Gemma2-9B | 4.2180 | 3.8527 | 4.6095 | 3.6149 | 3.5155 | 3.9860 | 4.7394 |
| Llama-3.1-8B | 0.034 | 0.031 | 0.034 | 0.029 | 0.0295 | 0.0302 | 0.0356 |

Table 7: Reconstruction Loss when conducting experiments for SAE-Based Model Utilization Index (MUI) analysis using Llama and Gemma Models

## B.2 COROLLARY 1 FOR MODEL TRAINING

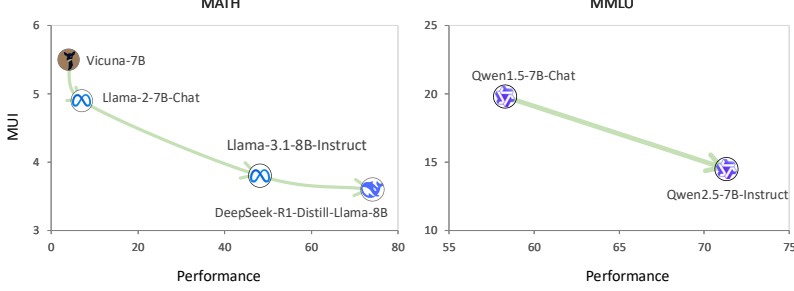

Figure 11: MUI (%) and Performance ACC (%) change trend compared between model Vicuna-7B, Llama-2-7B-Chat, Llama-3.1-8B-Instruct, and DeepSeek-R1-Distill-Llama-8, as well as between Qwen1.5-7B-Chat and Qwen2.5-7B-Instruct on task MATH and MMLU. The continual improvements in capabilities is reflected by a consistent decrease in MUI and an increase in performance.

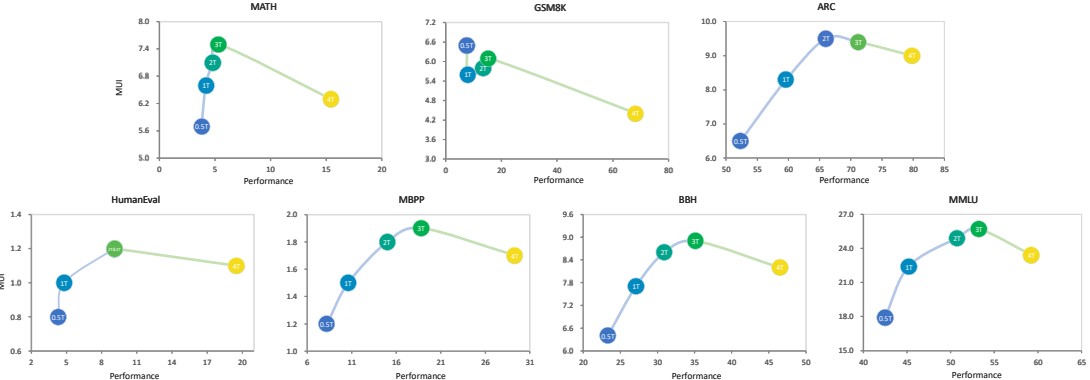

Figure 12: Performance (accuracy %) and Model Utilization Index (MUI) (%) of the OLMo series model across the Selected Seven Tasks.

| Model | GSM8K (Math & Reasoning) | MATH (Math & Reasoning) | ARC$_c$ (Math & Reasoning) | HumanEval (Code) | MBPP (Code) | BBH (General) | MMLU (General) |
|---|---|---|---|---|---|---|---|
| OLMo-2-7B-0.5T | 7.6 / 6.5 | 3.8 / 5.7 | 52.3 / 6.5 | 4.3 / 0.8 | 8.2 / 1.2 | 23.3 / 6.4 | 42.5 / 17.9 |
| OLMo-2-7B-1T | 8.0 / 5.6 | 4.2 / 6.6 | 59.6 / 8.3 | 4.8 / 1.0 | 10.6 / 1.5 | 27.1 / 7.7 | 45.2 / 22.4 |
| OLMo-2-7B-1.5T | 10.8 / 6.4 | 4.7 / 8.5 | 62.2 / 9.2 | 9.7 / 1.1 | 14.8 / 1.7 | 27.3 / 8.9 | 49.1 / 24.7 |
| OLMo-2-7B-2T | 13.5 / 5.8 | 4.8 / 7.1 | 66.0 / 9.5 | 9.1 / 1.2 | 15.0 / 1.8 | 30.9 / 8.6 | 50.7 / 24.9 |
| OLMo-2-7B-2.5T | 15.2 / 6.6 | 5.5 / 7.1 | 66.0 / 9.5 | 6.7 / 1.2 | 15.8 / 1.9 | 35.0 / 9.2 | 51.2 / 24.5 |
| OLMo-2-7B-3T | 15.3 / 6.1 | 5.3 / 7.5 | 71.2 / 9.4 | 9.1 / 1.2 | 18.8 / 1.9 | 35.1 / 8.9 | 53.2 / 25.7 |
| OLMo-2-7B-3.5T | 16.8 / 5.4 | 5.7 / 6.5 | 71.4 / 8.8 | 8.5 / 1.1 | 20.8 / 1.7 | 38.5 / 8.2 | 53.7 / 23.8 |
| OLMo-2-7B-4T (final) | 68.2 / 4.4 | 15.4 / 6.3 | 79.8 / 9.0 | 19.5 / 1.1 | 29.3 / 1.7 | 46.5 / 8.2 | 59.2 / 23.4 |

Table 8: Performance (accuracy %) under few-shot inference setting / Model Utilization Index (MUI) (%) of the OLMo series model. The detailed checkpoint information is shown in Appendix A.3.

| Model | GSM8K (Math & Reasoning) | MATH (Math & Reasoning) | ARC$_c$ (Math & Reasoning) | HumanEval (Code) | MBPP (Code) | BBH (General) | MMLU (General) |
|---|---|---|---|---|---|---|---|
| Llama-2-7B-Chat | 25.8 / 3.8 | 5.4 / 4.9 | 60.0 / 5.4 | 16.4 / 0.9 | 24.8 / 1.4 | 26.2 / 8.5 | 48.2 / 15.8 |
| CodeLlama-7B-Instruct | 23.0 / 4.5 | 6.0 / 7.0 | 49.5 / 8.7 | 34.1 / 1.8 | 44.7 / 2.9 | 21.8 / 8.3 | 41.4 / 24.5 |
| Qwen2.5-7B-Instruct | 84.5 / 1.9 | 61.9 / 2.8 | 86.8 / 4.2 | 71.3 / 0.7 | 62.1 / 1.0 | 66.0 / 5.3 | 71.3 / 14.5 |
| Qwen2.5-Coder-7B-Instruct | 79.6 / 3.7 | 49.5 / 6.2 | 83.8 / 9.4 | 75.0 / 1.5 | 65.9 / 2.4 | 57.8 / 11.0 | 65.6 / 30.3 |
| Qwen2.5-Math-7B-Instruct | 92.1 / 1.9 | 76.7 / 3.3 | 66.3 / 3.8 | 53.0 / 0.7 | 47.9 / 1.1 | 29.1 / 2.9 | 51.2 / 8.5 |

Table 9: Performance (accuracy %) and Model Utilization Index (MUI) (%), as determined by neuron analysis for Llama-2-7B-Chat, CodeLlama-7B-Instruct, Qwen2.5-7B-Instruct, Qwen2.5-Coder-7B-Instruct, and Qwen2.5-Math-7B-Instruct.

## B.3 COROLLARY 2. DATA CONTAMINATION DETECTION

| Model | GSM8K (Math & Reasoning) | MATH (Math & Reasoning) | ARC$_c$ (Math & Reasoning) | HumanEval (Code) | MBPP (Code) | BBH (General) | MMLU (General) |
|---|---|---|---|---|---|---|---|
| Llama-2-7B-Chat | 25.8 / 3.8 | 5.4 / 4.9 | 60.0 / 5.4 | 16.4 / 0.9 | 24.8 / 1.4 | 26.2 / 8.5 | 48.2 / 15.8 |
| Llama-Code-Leakage | 22.2 / 3.7 | 5.2 / 4.5 | 60.7 / 4.8 | 35.4 / 2.2 | 49.7 / 1.2 | 23.6 / 7.7 | 48.1 / 14.0 |
| Llama-Math-Leakage | 39.0 / 4.2 | 8.2 / 6.2 | 49.5 / 7.6 | 5.5 / 1.0 | 8.6 / 1.6 | 25.2 / 6.4 | 43.9 / 23.8 |
| Llama-3.1-8B-Instruct | 86.5 / 2.7 | 48.1 / 3.8 | 81.9 / 3.8 | 62.2 / 1.2 | 57.9 / 2.2 | 65.4 / 7.7 | 64.4 / 14.5 |
| Llama3.1-Code-Leakage | 86.6 / 3.1 | 40.2/ 4.5 | 81.2 / 5.0 | 81.7 / 1.2 | 69.1 / 1.9 | 48.1 / 6.9 | 66.3 / 15.1 |
| Llama3.1-Math-Leakage | 89.8 / 3.0 | 47.4 / 4.1 | 81.6 / 4.7 | 54.9 / 1.3 | 50.5 / 2.2 | 48.9 / 6.3 | 66.1 / 14.4 |
| Qwen2.5-7B-Instruct | 84.5 / 1.9 | 61.9 / 2.8 | 86.8 / 4.2 | 71.3 / 0.7 | 62.1 / 1.0 | 66.0 / 5.3 | 71.3 / 14.5 |
| Qwen2.5-Code-Leakage | 78.6 / 2.0 | 42.5 / 2.8 | 88.9 / 3.5 | 85.4 / 1.9 | 73.5 / 1.1 | 56.8 / 5.3 | 56.8 / 11.7 |
| Qwen2.5-Math-Leakage | 98.1 / 2.4 | 84.6 / 3.2 | 57.4 / 3.5 | 2.2 / 1.3 | 1.8 / 0.8 | 31.4 / 4.9 | 44.7 / 11.1 |

Table 10: Performance (accuracy %) and Model Utilization Index (MUI) (%) for Llama-2-7B-Chat, Llama-3.1-8B-Instruction, and Qwen2.5-7B-Instruct, including their post-data contamination training performance on code-related tasks (HumanEval and MBPP) and math-related tasks (MATH and GSM8K). The detailed training setting is shown in Appendix A.7.

### B.4 COROLLARY 4. POSITIVE CORRELATION BETWEEN DATA DIVERSITY AND MUI

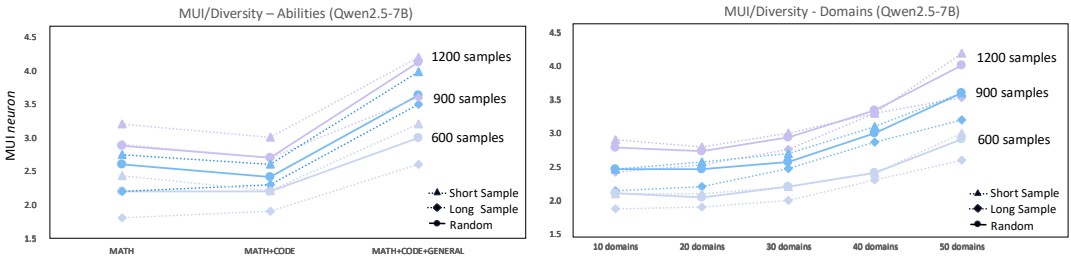

Figure 13: Using the Mann-Whitney U test, we evaluate the statistical significance of differences in MUI across different dataset sizes (600, 900, 1200) and text lengths (short vs. long) when using evaluation data testing different abilities. At a 95% confidence level, the analysis reveals that there are no statistically significant differences in proportions between short and long texts across the various dataset sizes. All p-values exceed the 0.05 threshold, indicating that text length does not significantly impact the MUI under these conditions.

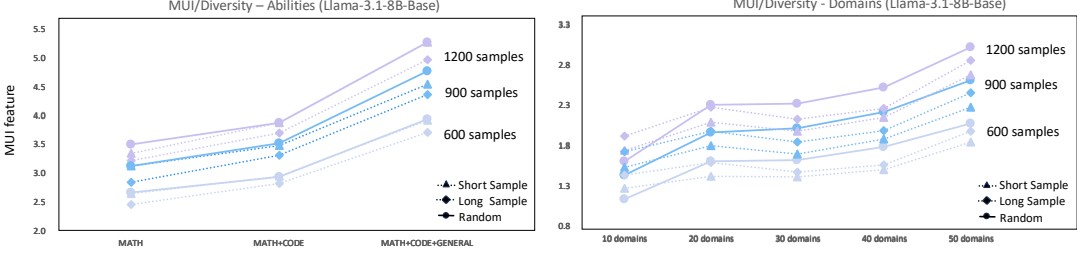

Figure 14: Using the Mann-Whitney U test, we evaluate the statistical significance of differences in MUI across different dataset sizes (600, 900, 1200) and text lengths (short vs. long) when using evaluation data testing different abilities. At a 95% confidence level, the analysis reveals that there are no statistically significant differences in proportions between short and long texts across the various dataset sizes. All p-values exceed the 0.05 threshold, indicating that text length does not significantly impact the MUI under these conditions.

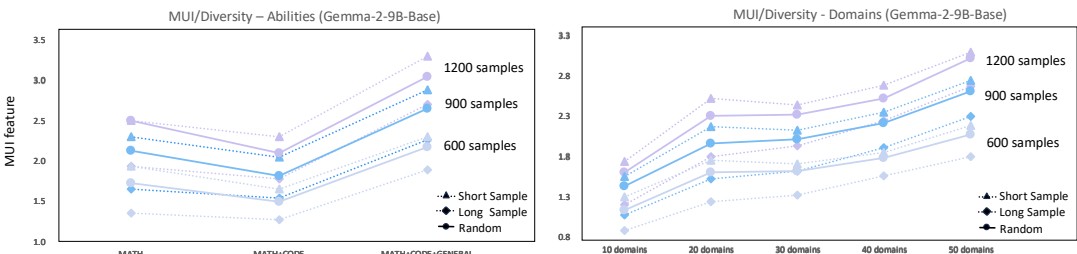

Figure 15: Using the Mann-Whitney U test, we evaluate the statistical significance of differences in MUI across different dataset sizes (600, 900, 1200) and text lengths (short vs. long) when using evaluation data testing different abilities. At a 95% confidence level, the analysis reveals that there are no statistically significant differences in proportions between short and long texts across the various dataset sizes. All p-values exceed the 0.05 threshold, indicating that text length does not significantly impact the MUI under these conditions.

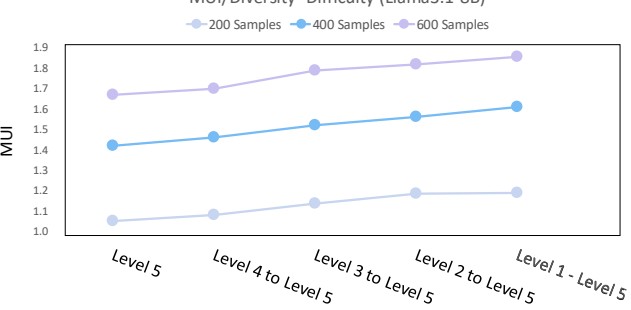

Figure 16: MUI (%) across various dataset sizes (200, 400, 600) and question difficulty levels evaluated on the MATH dataset. "Level x to Level y" denotes a uniform distribution of questions ranging from difficulty level x to y. The results suggest that merely increasing the difficulty does not enhance the MUI — the MUI for the highest difficulty level (level 5) is lower than the MUI across questions equally from levels 1 through 5.

## B.5 ABLATION ON ANSWER CORRECTNESS

We have utilized MUI to assess model capabilities and data diversity. However, there remains a significant issue to address to further validate the reliability of this metric — since MUI measures the activated capabilities, will it be affected by response quality? i.e., if only correct responses should be considered as valid instances of capability utilization. We conduct an ablation study on MUI by sampling an equal number of correct and incorrect responses. The results are shown in Figure 17 and Figure 18. Using the Mann-Whitney U test, we find that there is no significant difference in MUI between correct and incorrect samples, demonstrating the reliability of MUI, no matter models can provide correct responses or not.

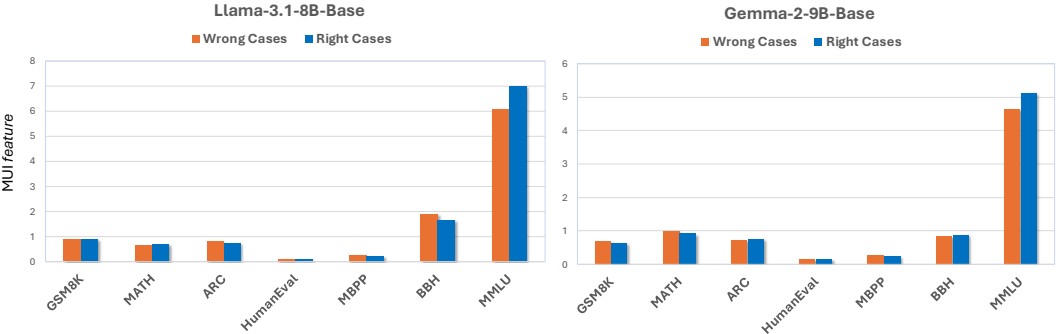

Figure 17: Ablation Study: the $\mathbf{MUI}_{feature}$ for model Llama-3.1-8B-Base and Gemma-2-9B-Base, evaluated on correct and incorrect cases from the seven tasks. The Mann-Whitney U test indicates no significant differences between MUI on correct and incorrect cases for the two models.

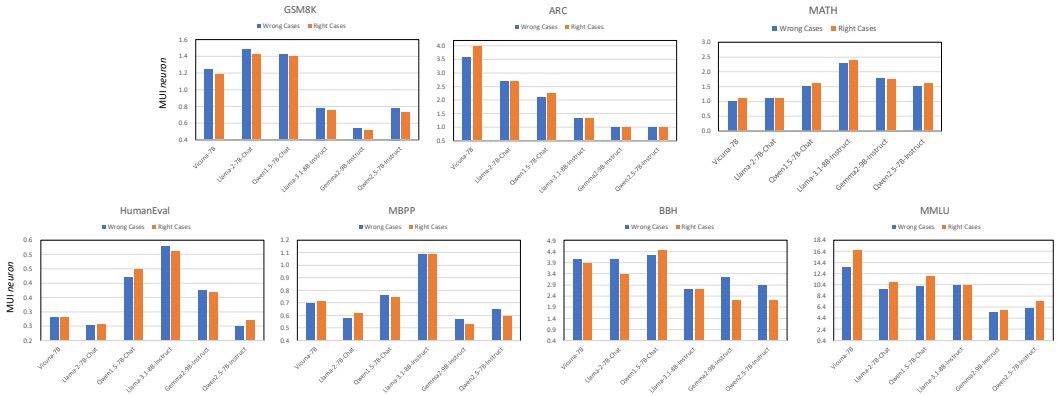

Figure 18: Ablation Study: the $\mathbf{MUI}_{neuron}$ for seven model evaluated on correct and incorrect cases from the seven tasks (more detailed experiment setting is shown in Section 3.6. The Mann-Whitney U test indicates no significant differences between MUI on correct and incorrect cases for the two models.

## B.6 IMPACTS OF DIFFERENT NEURON INTERPRETATION METHODS

Through utilizing the proposed two explicit interpretation methods, we have unveiled the Utility Law and four corollaries, demonstrating the feasibility of using MUI for model and data evaluation. Moving forward, we will not confine our analysis to a single, fixed neuron or SAEs analysis method. Instead, we aim to employ multiple techniques to ensure a comprehensive and robust analysis, adapting to advancements in the field of interpretability. In this section, we plan to incorporate a broader range of interpretation methods, which will include various definitions of neuron importance (con-

tribution score) and the application of different threshold, to conduct a broad MUI analysis (given that SAE analysis tends to be more static, our exploration for SAE interpretation primarily focus on expanding threshold choices).

### B.6.1 DETAILS OF OUR NEURON CONTRIBUTION SCORES USING LOGIT-LENS UNDER SWIGLU

We consider the feed-forward network (FFN) block with SwiGLU activation in a transformer layer $l$. Here $t$ indexes the position in the input sequence. Let the hidden input to the FFN be $\mathbf{x}_t^l \in R^{d_{\text{model}}}$. The SwiGLU block is parameterized by three projection matrices:

$$\mathbf{W}_g^l, \; \mathbf{W}_{\text{in}}^l \in R^{d_{\text{model}} \times d_{\text{int}}}, \quad \mathbf{W}_{\text{out}}^l \in R^{d_{\text{int}} \times d_{\text{model}}}.$$

**(1) Intermediate activation.** The intermediate activation of SwiGLU is

$$\mathbf{O}_t^l = \text{SiLU}\big(\mathbf{x}_t^l \mathbf{W}_g^l\big) \odot \big(\mathbf{x}_t^l \mathbf{W}_{\text{in}}^l\big) \; \in \; R^{d_{\text{int}}}. \tag{13}$$

**(2) FFN output.** The FFN output mapped back to the model dimension is

$$m_t^l = \mathbf{O}_t^l \mathbf{W}_{\text{out}}^l \; \in \; R^{d_{\text{model}}}. \tag{14}$$

**(3) Hidden state recursion.** With residual connections and attention outputs $a_t^l$, the hidden state at layer $l$ is

$$h_t^l = h_t^{l-1} + a_t^l + m_t^l, \quad \Rightarrow \quad h_t^L = h_t^0 + \sum_{l=1}^{L} \big(a_t^l + m_t^l\big). \tag{15}$$

**(4) Logits.** Let $\mathbf{W}_u \in R^{V \times d_{\text{model}}}$ be the unembedding matrix. The unnormalized logits at position $t$ are

$$\ell_t = \mathbf{W}_u h_t^L = \mathbf{W}_u(h_t^0 + \sum_{l=1}^{L} \big(a_t^l + m_t^l\big)) = \mathbf{W}_u h_t^0 + \sum_{l=1}^{L} \Big(\mathbf{W}_u a_t^l + \mathbf{W}_u m_t^l\Big). \tag{16}$$

**(5) Substituting SwiGLU form.** Plugging in $m_t^l = \mathbf{O}_t^l \mathbf{W}_{\text{out}}^l$ and Eq. 13 gives

$$\mathbf{W}_u m_t^l = \mathbf{W}_u \mathbf{W}_{\text{out}}^l \Big[ \text{SiLU}(\mathbf{x}_t^l \mathbf{W}_g^l) \odot (\mathbf{x}_t^l \mathbf{W}_{\text{in}}^l) \Big]. \tag{17}$$

**(6) Final SwiGLU logit-lens equation.** Therefore, the logits for predicting the next token are

$$\boxed{\ell_t = \mathbf{W}_u h_t^0 + \sum_{l=1}^{L} \Big(\mathbf{W}_u a_t^l + \mathbf{W}_u \mathbf{W}_{\text{out}}^l \big[ \text{SiLU}(\mathbf{x}_t^l \mathbf{W}_g^l) \odot (\mathbf{x}_t^l \mathbf{W}_{\text{in}}^l) \big]\Big)} \tag{18}$$

and the predicted token is

$$y_{t+1} = \arg \max \ell_t. \tag{19}$$

**(7) Neuron Contribution Score (token-level).** For layer $l$ and position $t$, we define a gate-only token-level importance for the intermediate neuron $i$ (the $i$-th column in $(\mathbf{x}_t^l \mathbf{W}_{\text{in}}^l)$) as:

$$\boxed{f_{\text{neuron}}(i, l, y_{t+1} \mid y_{<t+1}) = \big(\mathbf{W}_u \mathbf{W}_{\text{out}}^l \, \text{SiLU}\big(\mathbf{x}_t^l \mathbf{W}_g^l\big)\big)_{y_{t+1}, i}} \tag{20}$$

### B.6.2 OTHER NEURON CONTRIBUTION SCORES:

Except for projecting activation statistics into the vocabulary, the activation of the inner state can also represent the attribute score of a neuron for predicting $\hat{y}$ given input $x$. Therefore, following previous

work (Meng et al., 2022), we propose defining the direct use of activation as the contribution score. The contribution of each neuron to the predicted output is calculated as follows:

$$f_{\text{neuron\_activation}}(i, l, \hat{y} \mid x) = \left( \sigma \left( \mathbf{W}_{\text{in}}^l \mathbf{x}_{-1}^l \right) \right)_{i, \hat{y}} \tag{21}$$

In addition to considering direct activation values, we also explore another definition of neuron importance. Following the work (Dai et al., 2022), which utilizes Integrated Gradients to define the contribution of each neuron to predictions $\hat{y}$ given input $x$, we compute this contribution using the following formula:

$$f_{\text{neuron\_gradient}}(i, l, \hat{y} \mid x) = \int_{\alpha=0}^{1} \partial \left( \alpha \left( \sigma \left( \mathbf{W}_{\text{in}}^l \mathbf{x}_{-1}^l \right) \right)_{i, \hat{y}} \right) d\alpha \tag{22}$$

To approximate the integral, we use a Riemann sum approach as described in (Dai et al., 2022). This approximation is defined as: $f_{\text{neuron\_gradient}}(i, l, \hat{y} \mid x) \approx \frac{1}{m} \sum_{k=1}^{m} \partial \left( \frac{k}{m} \left( \sigma \left( \mathbf{W}_{\text{in}}^l \mathbf{x}_{-1}^l \right) \right)_{i, \hat{y}} \right)$, where $m$ (we deploy $m = 10$) is the number of approximation steps. In addition to using a token-level score, we also explore utilizing a response-level score, which aggregates the scores of each token $\hat{y}$ for each neuron to attribute importance. This method could be implemented based on the prescribed importance calculations from previous equations (Equation 3, Equation 21, Equation 6, and Equation 22).

$$f_{\text{neuron\_sum}}(i, l, y \mid x) = \sum_{\hat{y}_j \in y} f_{\text{neuron}}(i, l, \hat{y}_j \mid x + \hat{y}_{<j}) \tag{23}$$

### B.6.3 THRESHOLD AND SELECTION STANDARDS:

MUI is designed to measure the fraction of task-activated features or neurons relative to the entire model, so the rule used to decide which neurons count as "activated" is crucial. Here we take neuron-based MUI as an example, SAE-based MUI adopts a similar strategy. As defined in Equation 3, we assign every neuron a contribution score that quantifies its influence on the output given the current input. Prior work (Meng et al., 2022; Dai et al., 2022; Zhao et al., 2024) has employed three main thresholding strategies to identify key neurons: 1) Layer-level top-k (topk): selecting the $k$ neurons with the highest contribution in each layer. 2) Global top-k (global_topk): selecting the top $k$ neurons across the entire network. 3) Top-score (topscore): in each layer, selecting neurons whose contribution exceeds a fixed multiple of that layer's maximum.

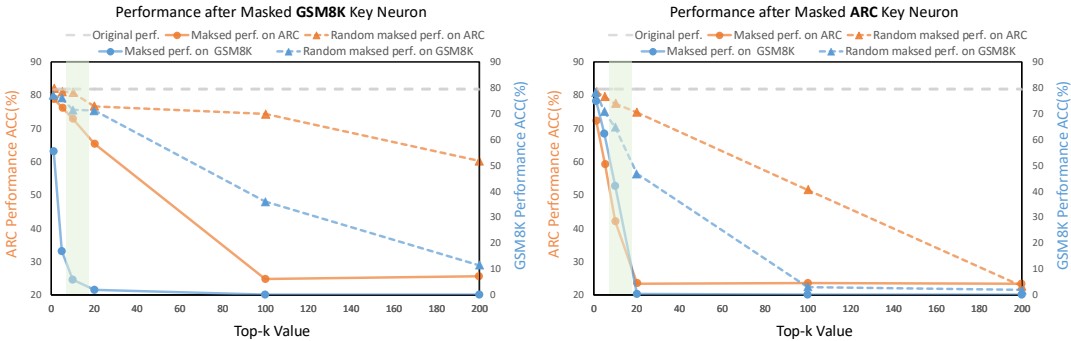

Figure 19: Performance of the Llama-3-8B-Instruct after the activated neurons being masked according to topk threshold. Left/right side selects activated neurons using ARC/GSM8K datasets. Our selection is based on the threshold marked in green box. Perf. is short for perfromance.

To keep MUI invariant to model size, we adopt a fixed ratio (i.e., the top one-thousandth) as the layer-level top-k threshold. Thus the numeric threshold varies across models and datasets. We validate this choice experimentally. Using Llama-3-8B-Instruct as base model and the ARC and

GSM8K datasets, we progressively mask the neurons selected at different $k$ values and present the resulting performance in Figures 19. Neuron masking is a widely used intervention technique in the field of mechanism interpretability that serves to verify whether a particular neuron bears a causal relationship to the model's output. Ideally, the more task-relevant key neurons that are masked, the more pronounced the resulting performance degradation. We can see that 1) Masking neurons identified by either GSM8K or ARC causes performance to fall steadily as more neurons are removed, and the decline is far steeper than when an equal number of neurons is masked at random, confirming that the selected neurons are indeed task-critical. 2) Specifically, when neurons derived from ARC are masked, the performance curves on both datasets (blue for GSM8K and orange for ARC in the left figure) are almost identical: when $k \in (0, 20)$, they both decline rapidly. In contrast, when neurons derived from GSM8K are masked (right figure), only the GSM8K performance drops sharply, whereas the ARC performance decreases much more gradually. This is because ARC encompasses, but is not limited to, mathematical reasoning, which is the primary focus of GSM8K. Hence, our threshold evidently falls within this interval, enabling us to separate neurons associated with distinct capabilities.

Based on this threshold selection standard, we test all possible defined importance measures of neurons, applying various threshold functions with differing threshold value. Our testing methods included: **1)** score-based importance $f_{\text{neuron}}$ (score) as detailed in Equation 3; **2)** activation-based importance $f_{\text{neuron\_activation}}$ (activate) as detailed in Equation 21; **3)** gradient-based importance $f_{\text{neuron\_gradient}}$ (gradient) as detailed in Equation 22; **4)** response-level score for $f_{\text{neuron}}$ (score_sum); **5)** response-level activation for $f_{\text{neuron\_activation}}$ (activate_sum). Each method are tested with a range of threshold functions with specific hyper-parameters: layer-level top-k (topk), global top-k (global_topk), and top-score (topscore). Details on other important measure combinations and threshold selections are provided in bellow.

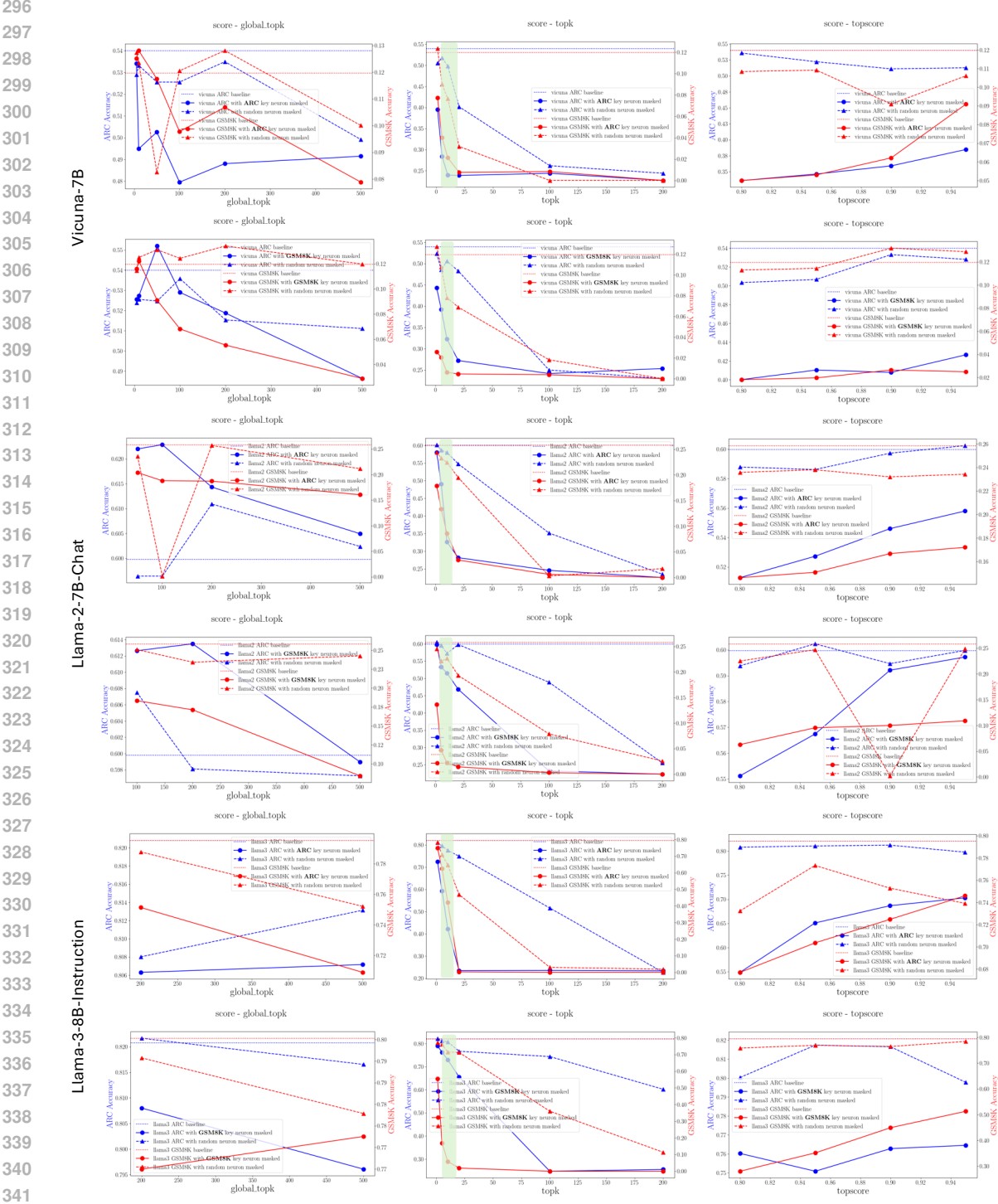

Figure 20: Performance accuracy (accuracy %) of the Vicuna-7B, Llama-2-7B-Chat, Llama-3-8B-Instruction model on the ARC and GSM8K datasets, with key neurons masked specifically for the **ARC** dataset or the **GSM8K** dataset. Key neurons are identified using a score-based importance measure (see Equation 3) and pre-defined threshold function (Detailed in Appendix A.5). The threshold value used for our MUI analysis —1‰, is visually indicated by a green box . The performance impact of masking an equivalent number of key neurons as in the **ARC / GSM8K** dataset on the corresponding model is represented with a dashed line.

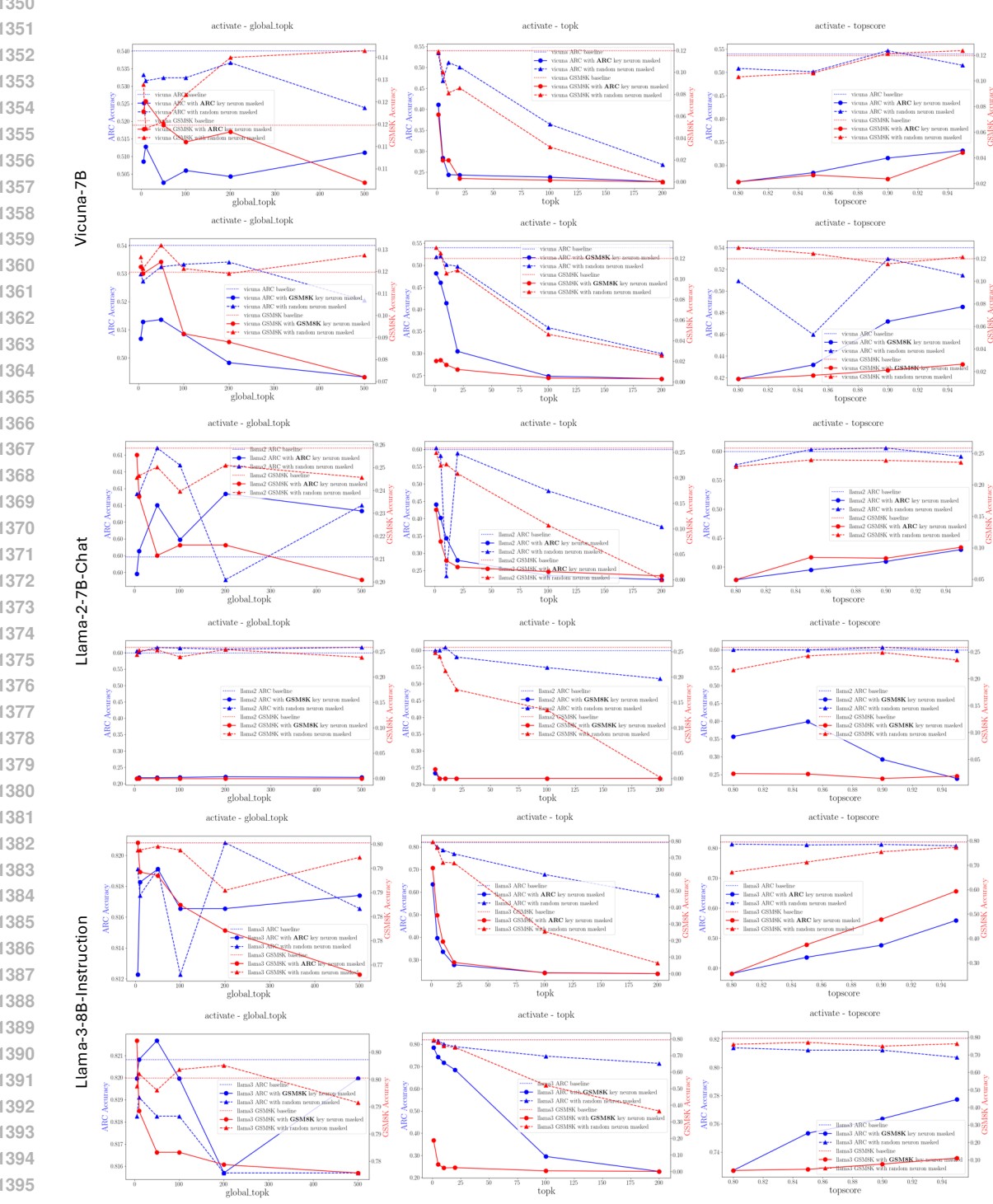

Figure 21: Performance (accuracy %) of the Vicuna-7B, Llama-2-7B-Chat, and Llama-3-8B-Instruction model on the ARC and GSM8K datasets, with key neurons masked specifically for the **ARC** dataset or the **GSM8K** dataset. Key neurons are identified using a score-based importance measure (see Equation 21) and pre-defined threshold function (Detailed in Appendix A.5). The performance impact of masking an equivalent number of key neurons as in the **ARC / GSM8K** dataset on the corresponding model is represented with a dashed line.

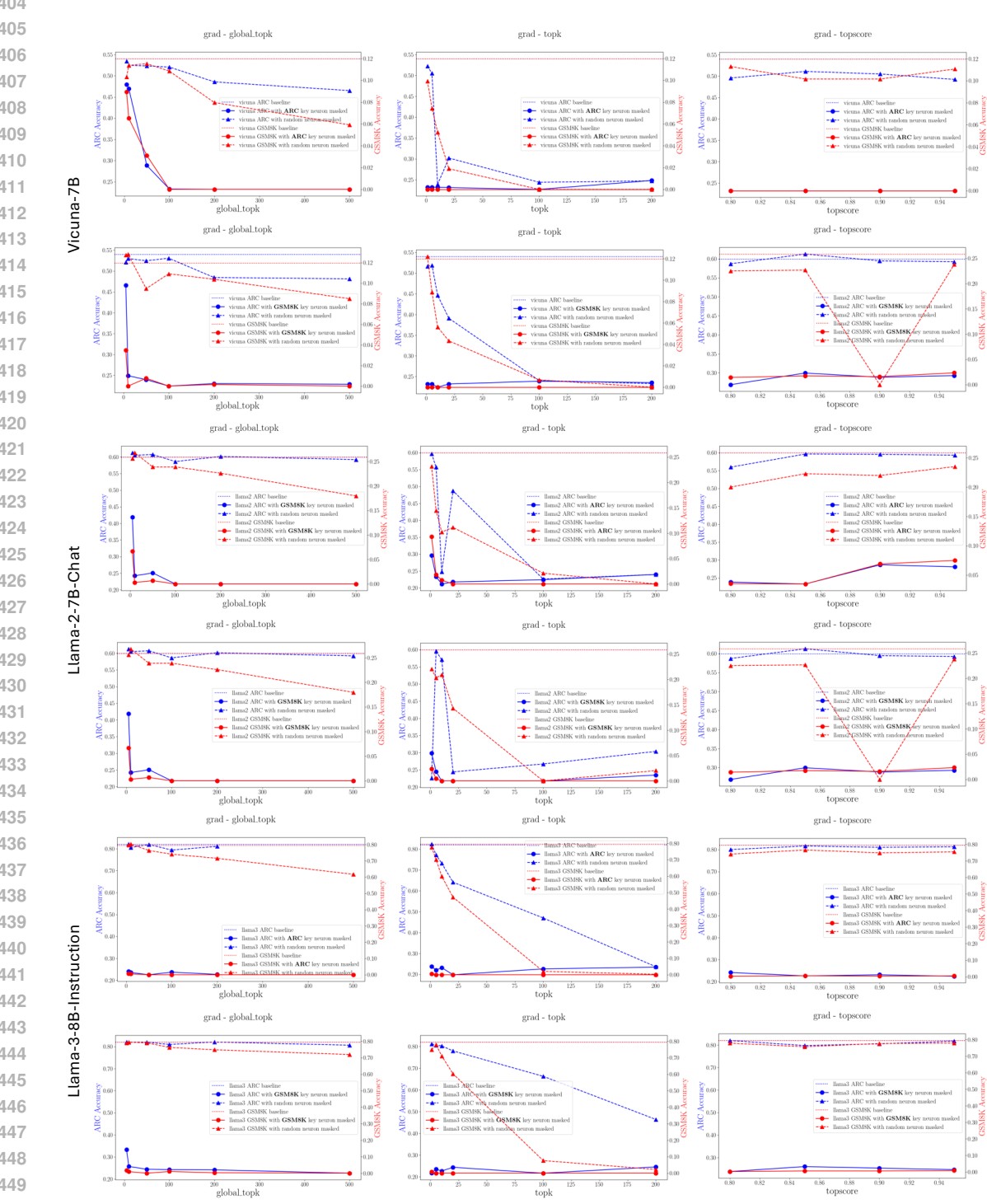

Figure 22: Performance (accuracy %) of the Vicuna-7B, Llama-2-7B-Chat, and Llama-3-8B-Instruction model on the ARC and GSM8K datasets, with key neurons masked specifically for the **ARC** dataset or the **GSM8K** dataset. Key neurons are identified using a score-based importance measure (see Equation 22) and pre-defined threshold function (Detailed in Appendix A.5). The performance impact of masking an equivalent number of key neurons as in the **ARC / GSM8K** dataset on the corresponding model is represented with a dashed line.

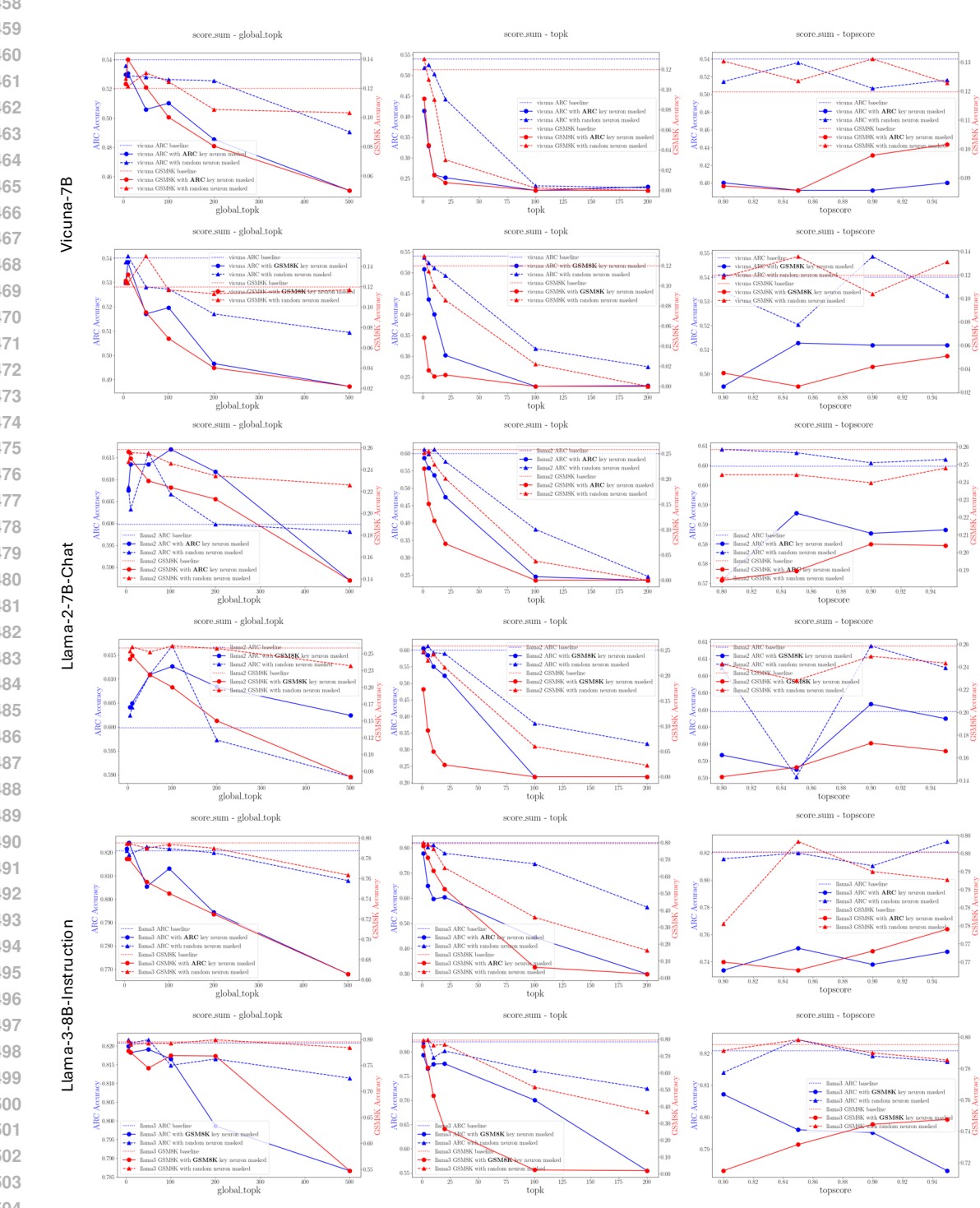

Figure 23: Performance (accuracy %) of the Vicuna-7B, Llama-2-7B-Chat, and Llama-3-8B-Instruction model on the ARC and GSM8K datasets, with key neurons masked specifically for the **ARC** dataset or the **GSM8K** dataset. Key neurons are identified using a score-based importance measure (see Equation 23) and pre-defined threshold function (Detailed in Appendix A.5). The performance impact of masking an equivalent number of key neurons as in the **ARC / GSM8K** dataset on the corresponding model is represented with a dashed line.

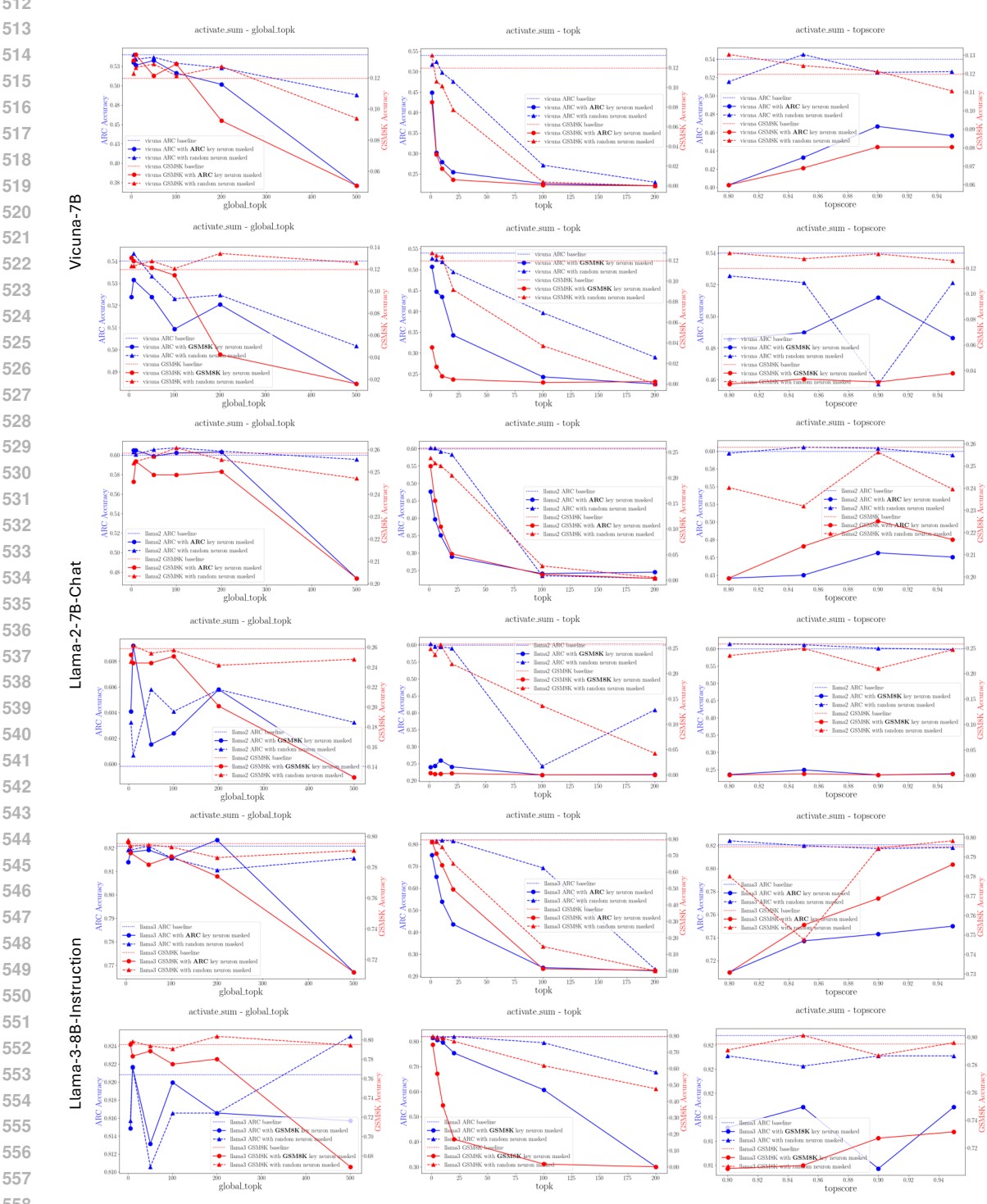

Figure 24: Performance (accuracy %) of the Vicuna-7B, Llama-2-7B-Chat, and Llama-3-8B-Instruction model on the ARC and GSM8K datasets, with key neurons masked specifically for the **ARC** dataset or the **GSM8K** dataset. Key neurons are identified using a score-based importance measure (see Equation 23) and pre-defined threshold function (Detailed in Appendix A.5). The performance impact of masking an equivalent number of key neurons as in the**ARC / GSM8K** dataset on the corresponding model is represented with a dashed line.

### B.6.4 IMPACT OF DIFFERENT NEURON INTERPRETATION METHODS

To demonstrate that our analysis is both generalizable and inclusive, we expand to incorporate various interpretation methods. Continuing from our previous threshold selections B.6.4, we conduct analyses using these methods to evaluate changes in the Model Utilization Index (MUI), with a particular focus on the consistency of MUI trends for different model. For some threshold values that do not integrate effectively, we randomly selected a threshold value within the testing range to facilitate a quantitative comparison. The results, which encompass the full parameter range, are detailed in the Figure 25. After applying different interpretation methods, we found that the MUI derived from various approaches (Table 13 and Table 14) correlates well with our experiment selections (Table 11), achieving an average Spearman correlation coefficient of 0.723. This indicates that the overall trend is relatively consistent that model capability and MUI is in a reverse Relationship. However, due to differences in the definition of importance, some models, such as Qwen2.5, exhibit a higher MUI when using activation-based methods compared to Llama3.1.

For SAE interpretation, where feature contribution scores are defined consistently, we considered various thresholds to examine consistency. The results revealed that the selected thresholds (Table 16 and Table 17) align closely with our experimental (Table 15) thresholds, achieving a Spearman correlation coefficient of 0.984. This high correlation demonstrates a strong consistency across different thresholds and confirms the reliability of our SAE interpretation approach in MUI analysis.

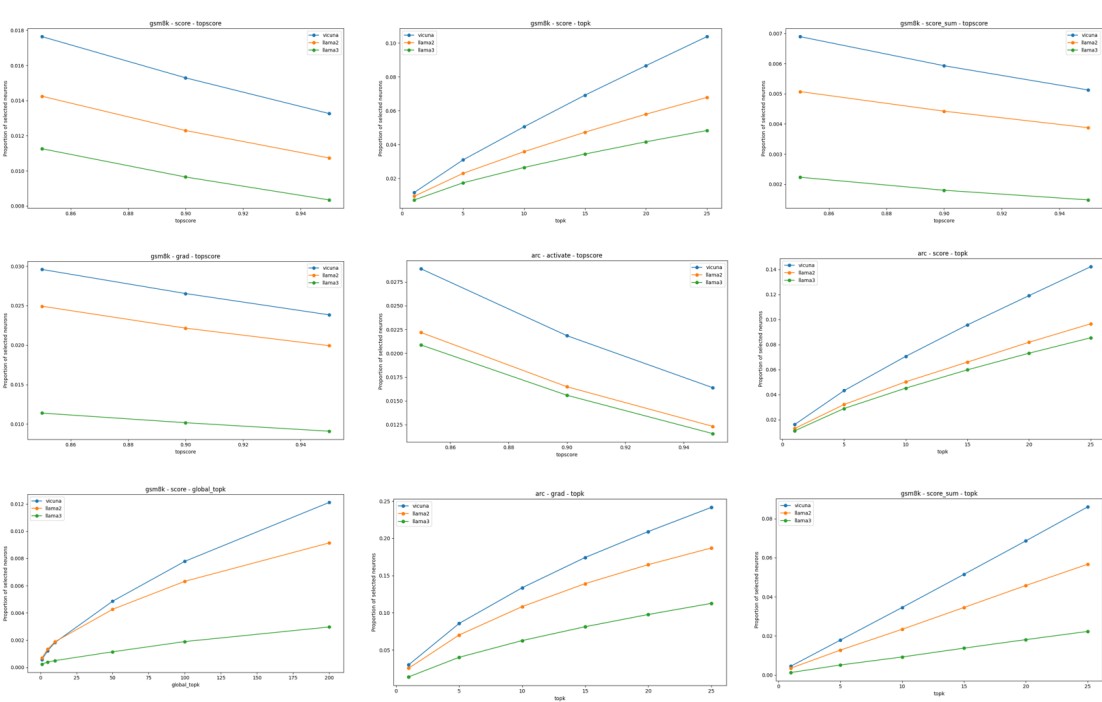

Figure 25: MUI (%) for Vicuna-7B, Llama-2-7B-Chat, and Llama-3-8B-Instruction, across various combinations of neuron importance definitions and threshold. The results demonstrate that the rank order of the three models remains consistent across different interpretation settings.

| Model | GSM8K (Math & Reasoning) | MATH (Math & Reasoning) | ARC$_c$ (Math & Reasoning) | HumanEval (Code) | MBPP (Code) | BBH (General) | MMLU (General) |
|---|---|---|---|---|---|---|---|
| Vicuna-7B | 11.9 / 5.0 | 4.0 / 5.1 | 54.0 / 7.1 | 13.4 / 1.1 | 23.6 / 1.7 | 28.6 / 7.9 | 45.3 / 24.4 |
| Llama-2-7B-Chat | 25.8 / 3.6 | 5.4 / 4.6 | 60.0 / 5.0 | 16.4 / 0.9 | 24.8 / 1.3 | 26.2 / 7.9 | 48.2 / 14.9 |
| Qwen1.5-7B-Chat | 58.7 / 2.8 | 17.9 / 3.4 | 76.2 / 5.7 | 39.0 / 0.9 | 39.8 / 1.4 | 41.3 / 8.0 | 58.3 / 18.6 |
| Llama-3.1-8B-Instruct | 86.5 / 2.2 | 48.1 / 3.0 | 81.9 / 2.4 | 62.2 / 0.9 | 57.9 / 1.7 | 63.4 / 6.1 | 64.4 / 11.8 |
| Qwen2.5-7B-Instruct | 84.5 / 1.3 | 61.9 / 2.1 | 86.8 / 2.8 | 71.3 / 0.5 | 62.1 / 0.8 | 61.5 / 3.7 | 71.3 / 10.2 |

Table 11: Performance acc(%) / Activated proportion MUI(%) of the selected model. With score-based importance (Equation 3) and top 1‰ threshold.

| Model | GSM8K (Math & Reasoning) | MATH (Math & Reasoning) | ARC$_c$ (Math & Reasoning) | HumanEval (Code) | MBPP (Code) | BBH (General) | MMLU (General) |
|---|---|---|---|---|---|---|---|
| Vicuna-7B | 11.9 / 6.9 | 4.0 / 6.8 | 54.0 / 9.6 | 13.4 / 1.5 | 23.6 / 2.2 | 28.6 / 10.7 | 45.3 / 32.4 |
| Llama-2-7B-Chat | 25.8 / 4.7 | 5.4 / 6.0 | 60.0 / 6.6 | 16.4 / 1.2 | 24.8 / 1.8 | 26.2 / 10.5 | 48.2 / 19.2 |
| Qwen1.5-7B-Chat | 58.7 / 3.7 | 17.9 / 4.5 | 76.2 / 7.9 | 39.0 / 1.2 | 39.8 / 1.9 | 41.3 / 10.8 | 58.3 / 24.3 |
| Llama-3.1-8B-Instruct | 86.5 / 2.9 | 48.1 / 4.0 | 81.9 / 4.0 | 62.2 / 1.3 | 57.9 / 2.3 | 63.4 / 8.0 | 64.4 / 15.2 |
| Qwen2.5-7B-Instruct | 84.5 / 1.7 | 61.9 / 2.6 | 86.8 / 3.7 | 71.3 / 0.7 | 62.1 / 1.1 | 61.5 / 4.7 | 71.3 / 13.0 |

Table 12: Performance acc(%) / Activated proportion MUI(%) of the selected model. With score-based importance (Equation 3) and topk ($k = 15$) threshold.

| Model | GSM8K (Math & Reasoning) | MATH (Math & Reasoning) | ARC$_c$ (Math & Reasoning) | HumanEval (Code) | MBPP (Code) | BBH (General) | MMLU (General) |
|---|---|---|---|---|---|---|---|
| Vicuna-7B | 11.9 / 3.2 | 4.0 / 3.5 | 54.0 / 5.0 | 13.4 / 0.8 | 23.6 / 1.1 | 28.6 / 4.7 | 45.3 / 15.3 |
| Llama-2-7B-Chat | 25.8 / 2.5 | 5.4 / 3.4 | 60.0 / 3.9 | 16.4 / 7.3 | 24.8 / 1.1 | 26.2 / 5.5 | 48.2 / 11.2 |
| Qwen1.5-7B-Chat | 58.7 / 2.4 | 17.9 / 2.6 | 76.2 / 4.4 | 39.0 / 7.3 | 39.8 / 1.0 | 41.3 / 5.6 | 58.3 / 13.1 |
| Llama-3.1-8B-Instruct | 86.5 / 1.7 | 48.1 / 2.4 | 81.9 / 2.5 | 62.2 / 7.5 | 57.9 / 1.2 | 63.4 / 4.1 | 64.4 / 9.1 |
| Qwen2.5-7B-Instruct | 84.5 / 2.0 | 61.9 / 3.5 | 86.8 / 4.8 | 71.3 / 9.6 | 62.1 / 1.6 | 61.5 / 6.0 | 71.3 / 19.6 |

Table 13: Performance acc(%) / Activated proportion MUI(%) of the selected model. With activation-based importance (Equation 21) and topk ($k = 10$) threshold.

| Model | GSM8K (Math & Reasoning) | MATH (Math & Reasoning) | ARC$_c$ (Math & Reasoning) | HumanEval (Code) | MBPP (Code) | BBH (General) | MMLU (General) |
|---|---|---|---|---|---|---|---|
| Vicuna-7B | 11.9 / 2.4 | 4.0 / 4.7 | 54.0 / 3.4 | 13.4 / 0.6 | 23.6 / 1.0 | 28.6 / 3.4 | 45.3 / 14.1 |
| Llama-2-7B-Chat | 25.8 / 2.0 | 5.4 / 3.9 | 60.0 / 2.9 | 16.4 / 0.4 | 24.8 / 0.7 | 26.2 / 3.8 | 48.2 / 11.4 |
| Qwen1.5-7B-Chat | 58.7 / 1.4 | 17.9 / 3.0 | 76.2 / 2.4 | 39.0 / 0.3 | 39.8 / 0.7 | 41.3 / 3.0 | 58.3 / 9.4 |
| Llama-3.1-8B-Instruct | 86.5 / 0.5 | 48.1 / 1.5 | 81.9 / 0.8 | 62.2 / 0.2 | 57.9 / 0.5 | 63.4 / 1.4 | 64.4 / 4.4 |
| Qwen2.5-7B-Instruct | 84.5 / 1.7 | 61.9 / 2.5 | 86.8 / 2.2 | 71.3 / 0.2 | 62.1 / 0.5 | 61.5 / 2.3 | 71.3 / 16.4 |

Table 14: Performance acc(%) / Activated proportion MUI(%) of the selected model. With activation-based importance (Equation 22) and top score ($k = 0.95$) as the threshold.

| Model | GSM8K (Math & Reasoning) | MATH (Math & Reasoning) | ARC$_c$ (Math & Reasoning) | HumanEval (Code) | MBPP (Code) | BBH (General) | MMLU (General) |
|---|---|---|---|---|---|---|---|
| Llama-3.1-8B | 42.9 / 1.9 | 15.1 / 2.0 | 77.1 / 3.2 | 19.5 / 0.40 | 26.1 / 0.7 | 50.1 / 3.9 | 59.6 / 13.0 |
| Gemma2-9B | 65.8 / 1.7 | 20.8 / 2.4 | 84.6 / 3.2 | 54.8 / 0.37 | 57.7 / 0.5 | 67.4 / 2.2 | 67.2 / 9.6 |

Table 15: Performance (accuracy %) under one-shot inference setting / SAE-based Model Utilization Index (MUI) (%) of the selected model. With topk (k=50) as the threshold.

| Model | GSM8K (Math & Reasoning) | MATH (Math & Reasoning) | ARC$_c$ (Math & Reasoning) | HumanEval (Code) | MBPP (Code) | BBH (General) | MMLU (General) |
|---|---|---|---|---|---|---|---|
| Llama-3.1-8B | 42.9 / 0.6 | 15.1 / 0.7 | 77.1 / 1.0 | 19.5 / 0.10 | 26.1 / 0.2 | 50.1 / 1.4 | 59.6 / 5.6 |
| Gemma2-9B | 65.8 / 0.5 | 20.8 / 0.7 | 84.6 / 0.9 | 54.8 / 0.08 | 57.7 / 0.1 | 67.4 / 0.7 | 67.2 / 4.8 |

Table 16: Performance (accuracy %) under one-shot inference setting / SAE-based Model Utilization Index (MUI) (%) of the selected model. With topk (k=10) as the threshold.

| Model | GSM8K (Math & Reasoning) | MATH (Math & Reasoning) | ARC$_c$ (Math & Reasoning) | HumanEval (Code) | MBPP (Code) | BBH (General) | MMLU (General) |
|---|---|---|---|---|---|---|---|
| Llama-3.1-8B | 42.9 / 0.24 | 15.1 / 0.31 | 77.1 / 0.44 | 19.5 / 0.05 | 26.1 / 0.09 | 50.1 / 0.62 | 59.6 / 2.89 |
| Gemma2-9B | 65.8 / 0.21 | 20.8 / 0.32 | 84.6 / 0.40 | 54.8 / 0.03 | 57.7 / 0.04 | 67.4 / 0.31 | 67.2 / 1.79 |

Table 17: Performance (accuracy %) under one-shot inference setting / SAE-based Model Utilization Index (MUI) (%) of the selected model. With top score ($k = 0.9$) as the threshold.

## C  FEW-SHOT INSTRUCTION

---

**ARC:**
Question: A comet passes close to a planet, causing a slight wobble in its orbit. Which is the most likely effect of this wobble?
A. The planet will move faster in its orbit.
B. The planet's orbit will become more elliptical.
C. The planet's axial tilt will change.
D. The planet's temperature will increase.

Answer: The correct answer is C. The planet's axial tilt will change. The wobble in the orbit, also known as "axial precession," can alter the angle at which the planet tilts on its axis, affecting its seasons and climate.

---

**GSM8K:**
Question: Weng earns $12 an hour for babysitting. Yesterday, she just did 50 minutes of babysitting. How much did she earn?
Answer: Weng babysat for 50 minutes, which is $\frac{50}{60} = \frac{5}{6}$ of an hour. At $12 per hour, her earnings are:
$$12 \times \frac{5}{6} = 10 \text{ dollars.}$$
So the answer is $10.

---

**HumanEval & MBPP:**

Question: Write a Python function `alternate(numbers:  List[int], value: int) -> List[int]` to solve the following problem: Alternate between the elements of the input list `numbers` and the `value`. The `value` should appear between every two consecutive elements of `numbers`, and at the beginning and end.

```
>>> alternate([], 5)
[]
>>> alternate([1, 2, 3], 5)
[5, 1, 5, 2, 5, 3, 5]
```

Answer: To implement the `alternate` function, we can iterate through the list and insert the value between each element of the list, as well as at the beginning and end of the list. Here's how you can do it:

```python
from typing import List

def alternate(numbers: List[int], value: int) -> List[int]:
    """Alternate between the elements of the input list `numbers`
        ↪ and the `value`.
    >>> alternate([], 5)
    []
    >>> alternate([1, 2, 3], 5)
    [5, 1, 5, 2, 5, 3, 5]
    """
    if not numbers:  # If the list is empty, return an empty list
        return []

    result = [value]  # Start with the value at the beginning
    for num in numbers:
        result.append(num)
        result.append(value)  # Add value after each number
    return result
```

**MATH:**

Problem: Find $\tan 45°$.

Solution: We know that $\tan 45° = \frac{\sin 45°}{\cos 45°}$. Since $\sin 45°$ and $\cos 45°$ are both equal to $\frac{\sqrt{2}}{2}$, it follows that:

$$\tan 45° = \frac{\sin 45°}{\cos 45°} = \frac{\frac{\sqrt{2}}{2}}{\frac{\sqrt{2}}{2}} = 1.$$

Therefore, the solution is 1.

**MMLU:**

Question:Where is the sinoatrial node located?
A. Between the left atrium and the left ventricle
B. Between the right atrium and the right ventricle
C. In the upper wall of the right atrium
D. In the upper wall of the left ventricle

Answer: The correct answer is C.
Here's why: The sinoatrial (SA) node is the heart's natural pacemaker. It's responsible for initiating the electrical impulses that cause the heart to contract and pump blood.Its location in the right atrium allows it to efficiently send these impulses throughout the heart.

