# OpenReview forum: "Model Utility Law: Evaluating LLMs beyond Performance via Mechanistically Interpretable Metric"
_ICLR.cc/2026/Conference — Submitted to ICLR 2026_

### Official Review · Reviewer_m7CX · 2025-10-18

**Soundness:** 2
**Presentation:** 2
**Contribution:** 1
**Rating:** 2
**Confidence:** 4

**Summary:**

This paper introduces the model utilization index (MUI). The metric works as follows: given a task composed of instances numbered $t=1,..., T$, compute for each instance the logit lens (W_U[next_token] @ W_out[i] * act_i) of each active neuron ($i$) at each position, with respect to the next token predicted at that position. Define $N(t)$ as the set of top-$k$ such neurons across all layers/positions for task instance $t$; this metric can also be used on SAEs, where $N(t)$ is chosen based on feature activation alone. The MUI is defined as the size of the union of $N(t)$ across all task instances $t$, divided by the number of neurons that are ever active on task $t$.

The authors then use MUI to study 4 phenomena: model training (MUI increases, then decreases, over the course of model pretraining.), data contamination (MUI increases when training on test), model comparison (they use MUI to "correct" purely performance-based metrics), and data diversity (MUI is higher on more diverse data). They conclude that MUI is useful across a wide range of scenarios.

**Strengths:**

This paper proposes a new metric, MUI, which I have not seen proposed elsewhere. It attempts this metric in a variety of scenarios (model training / comparison, data contamination / diversity), some more effectively than others. If effective, MUI might allow one to measure broader model capabilities

**Weaknesses:**

**Unclear experimental details / setup**: Some details of the experiments are not clear. I have put these in the Questions section.

**Flawed evaluation**: The authors rightly claim that
> Given the massive scale of pre-training data, it is nearly impossible to determine which samples a model has previously seen and which it has not. These limitations call for a new evaluation paradigm — how to estimate a model’s capabilities that may not be covered by the given limited testing data, namely generalizable evaluation

But when evaluating MUI, they evaluate on said existing, limited testing data, comparing MUI on that data to performance on that data. It's thus not clear to me how we can know if MUI serves as anything more than a proxy for performance on these tasks. It seems like you'd really need to measure MUI on one dataset and show that it generalizes to others?

**Limited utility**: Given the previous weakness, it's not clear when this metric would be useful. The ideal case would be to run it across a limited set of examples, and then extrapolate from that to models' task abilities? This way, you avoid evaluating the model on a larger set of tasks, which could be expensive. But, the relationship between MUI and individual task performance is not always stable: MUI and performance are less clearly linked on some tasks than others, namely HumanEval and MBPP. Indeed, model performance on each benchmark is actually different.

**Theoretical flaws**: For lack of a better explanation, MUI does not make very much sense. Taking lower MUI as a good thing, MUI seems to reward models whose top neurons (by logit lens) are similar across the different tasks in the task dataset. That is, the numerator is the union of N(t) across examples t, and this is minimized when the contents of each N(t) are similar. Since the size of each N(t) is a constant proportion of N_total, there's no real way to minimize N_total. The question is: why is it a good thing if the top neurons across each task example are similar? The authors claim that the MUI "reflects[s] the "effort" the model exerts to achieve a response", but I don't think that this interpretation, or the similar "MUI is the range of abilities activated" explanation holds water. Ultimately, the score given to each neuron is more or less its direct effect on the logits. This doesn't seem to be the same as effortfulness, and I don't think that these neurons reflect different abilities, so much as different tokens being upweighted, or different ways of upweighting these tokens. I'm surprised that the authors don't discuss memorization vs. generalization more, as I think this would be a more profitable angle of attack.

There is also another curious point: MUI uses the logit lens of neurons, but not the logit lens of SAE latents. It's not clear why the two should be treated differently.

**Only smaller / weaker models are studied**: This paper only studies models of 7-9B parameters, and entirely omits Gemma-3 and Qwen-3, even though the latter has an 8B model. This is a significant flaw: for this technique to be useful, we have to know that it works at scale. The technique that the authors propose does not seem terribly compute-intensive, meaning that it would not be very expensive to run this technique on larger models; I'm not sure why they say that doing MUI computation doubles the GPU-hours used - you can basically compute the neuron scores for each task example with 1 forward + 1 backward pass.

**Typos**: There are many typos and many grammatical errors; consider having this paper looked over. I stopped looking for these, because they are too numerous. The number of errors makes one worry about the quality of the paper.
- 015: advocate *for*
- 100: sparse autoencoder (not activation encoding); you could also just say sparse dictionary learning
- 191-3: HumanEval, ARC-Challenge; BigBench citation is wrong
- 198 Should be ~ (\sim)

**Questions:**

- $N_{neuron}(t)$ is defined with a constant $\eta$ over all positions / layers, but the text says "we pick up Top k% neurons with
highest activation values in each layer to setup the threshold $\eta$". Does this mean that $\eta$ is indeed constant? Or do you pick the top $k$% neurons from each layer?
- Do the token positions considered for MUI range over all tokens produced by the model in its response? Or just the tokens where the model is actually producing the answer? What about e.g. reasoning / chain of though tokens?
- Why does MUI use the logit lens of neurons, but not the logit lens of SAE latents?
- 392-393: You say "Therefore, we manually order nine base models as reference". These seems like bad practice. How do you do so?
- 3.6 (MUI FOR DATA DIVERSITY): Do you think this correlation with data diversity is specific to MUI? It seems like, insofar as more diverse data causes different neurons to activate / be produced by our models, it makes sense that MUI should increase. But this could be detected by measuring tokens / n-grams alone, or just neurons/SAE features active (but not MUI)

---

### Official Review · Reviewer_JNUR · 2025-10-21

**Soundness:** 2
**Presentation:** 1
**Contribution:** 1
**Rating:** 0
**Confidence:** 5

**Summary:**

This paper introduces the Model Utilization Index (MUI), a metric that measures the proportion of neurons or features activated during LLM inference to complement traditional performance scores. The metric is tested empirically across models and datasets, trying to establish patterns.

**Strengths:**

The topic of activation sparsity can be of interest if new results are shown. However, here is a report by perplexity to the query
"find papers that discuss the ratio of neurons that are truly active during transformer inference and try to link it to the model's ability."
(I sampled some key work from below, including "Lazy Neuron" and "and "Sparsing Law", and it is not cited )

Recent work highlights that only a small fraction of neurons in transformers are truly active during inference, and this sparsity appears to both influence and reflect a model’s generalization and interpretability capacity.

Key Empirical Findings
The seminal paper “The Lazy Neuron Phenomenon: On Emergence of Activation Sparsity in Transformers” by Z. Li et al. (Google Research, 2022) quantified neuron activity in models like T5 and ViT, reporting that typically only about 3–6% of neurons in transformer MLPs are active on average. Notably, larger models are sparser—as model width or depth increases, the ratio of nonzero activations decreases further. Around 93% of T5-base neurons activate less than 10% of the time, and no neurons are entirely “dead”, suggesting functional selectivity rather than redundancy.​

This sparsity manifests without explicit regularization and parallels biological neuron activity patterns. The study linked it to improved robustness, calibration, and noise tolerance, showing that applying artificial sparsification (via Top‑k masking) further enhanced these properties.​

Relation to Model Ability and Interpretability
A Stanford CS224N interpretability project (2022) examined activation sparsity as a mechanism promoting interpretable internal representations. It argued that sparse neuron firing helps isolate key features that drive predictions — analogous to disentangled feature learning — leading to better human interpretability of attention pathways without degrading task accuracy.​

Similarly, the 2024 “Sparsing Law” study formalized empirical scaling relations connecting LLM size to activation sparsity. It showed that more capable LLMs exhibit higher sparsity ratios, positing a “sparse efficiency law” linking sparsity to both efficiency and representational parsimony.​

Broader Theoretical and Dynamic Context
“Exploring the Benefit of Activation Sparsity in Pre-training” (Zhang et al., 2024) found that pretraining with inherent activation sparsity improved language understanding and reduced overfitting on downstream tasks.​

“Hidden Dynamics of Massive Activations in Transformer Models” (Gallego-Feliciano et al., 2025) studied rare but extremely high-valued activations (“massive activations”) and showed these few scalar outliers play a critical functional role, suggesting transformer performance depends disproportionately on a minority of active neurons.​

“Discovering Influential Neuron Paths in Vision Transformers” (2025) and “How Altering a Handful of Neurons Can Cripple Language Models” (2025) extended this notion, showing that a compact subset of neurons or paths carries the decisive signal flow through the network — confirming that only a small fraction of neurons determine model ability.

**Weaknesses:**

Very sloppy paper with many issues. Writing is unclear and the flow does not make sense. Even the abstract's first sentence is not related to what is shown in the rest of the abstract or the paper. Same for the first paragraph of the introduction -- it is unrelated to the next paragraph without major leaps. The writing is informal in many places, "expend less effort" "fundamental capability enhancement". Figure 1 does not really show anything. The sentences appear cut at places, and some statements are left hanging ("MUI is still limited by the current...").

For novelty, see "strengths"

The crucial question of determining the threshold is not discussed in the paper, as far as I can see.

Above all issues --- the results are underwhelming. If the message was as clear as "the more capable the model is on a sample, the lower the MUI" (or anything else), then the reader would benefit. However, there are multiple patterns, which lead the authors to come up with an elaborate system of four types. This way, they can support any observation. I see the results and I am left confused: why is MUI behaving similarly in and out of domain (accuracy of course increases in domain). Fig 5 shows that performance increases udring training and the MUI changes in some way. Is there a pattern we can learn from?

Table 1 uses a new score (combining sparsity with accuracy) with a parameter alpha (eq. 7) and then shows a small (?) improvement in correlation with a "manual order [of] nine base models as reference". The only way this table would be meaningful is if MUI would be used without the accuracy as an alternative, and the reference order would have some objectivity.

The experiment in Figure 8 basically says that the more varied the data, the more neurons are caught being active. (I'm informal, but again, I'm not even sure how the activity threshold is set).

**Questions:**

see above

---

### Official Review · Reviewer_9KXC · 2025-10-30

**Soundness:** 3
**Presentation:** 3
**Contribution:** 3
**Rating:** 6
**Confidence:** 3

**Summary:**

The paper proposes the Model Utility Law, a new framework for evaluating large language models (LLMs) beyond conventional performance metrics. It introduces the Model Utilization Index (MUI), which measures the fraction of neurons or features actively used when solving tasks, with the key idea that more capable models achieve higher performance while using fewer resources (lower MUI). The authors empirically show an inverse relationship between MUI and performance across multiple models and datasets, and demonstrate applications including detecting data contamination, comparing model efficiency, and monitoring training dynamics. MUI provides a mechanistically interpretable metric that complements standard accuracy measures, highlighting inefficiencies, memorisation tendencies, and generalisation capacity in LLMs.

**Strengths:**

1. The paper introduces the Model Utilization Index (MUI), a metric that quantifies the fraction of neurons or features actively used by a model when performing a task. Unlike traditional metrics such as accuracy or F1 score, MUI provides insight into how efficiently the model leverages its internal resources.
2. By linking performance to neuron or feature activation, the approach allows researchers to understand the internal mechanisms behind model decisions. This gives a transparent view of which parts of the network are contributing to task-solving, potentially revealing redundancy or inefficiency in the model.
3. The authors evaluate multiple LLM families (e.g., Llama‑2‑7B‑Chat, Qwen2.5‑7B) and various benchmark datasets (math, code, reasoning, general knowledge). The observed consistent inverse relationship between MUI and performance demonstrates that the metric is broadly applicable and meaningful.
4. MUI is shown to be useful for several downstream purposes: detecting data contamination or memorisation, comparing model efficiency beyond raw accuracy, and assessing whether performance gains are genuine or merely due to overfitting. This practical relevance strengthens its utility for both research and deployment.
5. By monitoring MUI over training, the authors classify model behaviour into phases, accumulating, evolving, coarsening, and collapsing. This provides a new lens for understanding learning dynamics and identifying points where training may become inefficient or unstable.

**Weaknesses:**

1. The accuracy and meaningfulness of MUI rely heavily on the method used to identify “key neurons” or “key features.” Different interpretability techniques (e.g., neuron attribution, feature sparsity) may yield different MUI values, limiting reproducibility and generality.
2. The observed inverse relationship between MUI and performance is empirical and correlational. The paper does not demonstrate that reducing MUI causes better performance or generalisation, so the metric is descriptive rather than causal.
3. MUI is an aggregate metric over neurons/features, which may obscure which specific layers or submodules are being inefficiently used. This could limit actionable insights for model improvement or pruning.
4. The paper focuses on standard benchmarks (math, code, reasoning). It does not explore MUI in settings like in-context learning, multimodal tasks, or generative tasks, where neuron usage dynamics may differ.

**Questions:**

1. How sensitive is MUI to the choice of interpretability method (e.g., neuron attribution vs sparse autoencoder features)?
2. Can you clarify whether lower MUI causes better generalisation, or is it purely a correlated phenomenon?
3. MUI is aggregated over all neurons/features. Can it be decomposed per layer or submodule to provide more actionable insights for model design or pruning?
4. can MUI be computed dynamically per token or per time step for generative tasks, and does it still maintain the inverse relationship with performance in such fine-grained settings?

---

### Official Review · Reviewer_nXEt · 2025-10-31

**Soundness:** 1
**Presentation:** 2
**Contribution:** 1
**Rating:** 2
**Confidence:** 4

**Summary:**

The paper introduces a new metric, the Model Utilization Index (MUI), which measures the proportion of neurons or features activated during inference. Experiments across nine LLMs reveal an inverse relationship between MUI and accuracy.  Some insights are interesting but not necessarily correct.

**Strengths:**

- The paper is generally easy to follow.
- Extensive experiments are conducted.

**Weaknesses:**

- The metric is built upon a vague concept and a wrong hypothesis, and thus lacks validity.
- - The motivation for the new metric is to better capture LLMs' capabilities. However, the paper does not define what those "true capabilities" are. You must at least specify what you are measuring. In fact, the authors abuse the notion of "capability". In Section 2.2, the authors use the number of "capabilities" to denote the number of neurons. I don't see where the equivalence comes from.
- - The authors assume that a stronger model should achieve higher performance while showing lower utilization. The paper neither explains why this hypothesis holds nor justifies it. One might argue that a stronger model should reach higher performance with lower compute, but the "lower utilization" defined here does not equate to compute, which is easy to measure.
- The experimental design is inappropriate. To validate the metric, the authors "manually order nine base models as reference," which is not a valid scientific approach and cannot justify the metric.

**Questions:**

- If I understand correctly, $W_u$ projects the final hidden layer output to vocabulary probabilities. Why does applying it to any hidden layer represent the contribution scores of neurons of the layer? This is far from "straightforward" in my opinion.

---

### Meta-Review · Area_Chair_c3EJ · 2026-01-07

**Summary:**

This paper proposes the Model Utilization Index as a means of assessing the differences between model performance on benchmarks and an underlying "true capability" level. As highlighted by the reviewers, the paper suffers from a severe lack of precision in the approach. The core idea is that models which use a smaller fraction of their capability on a set of tasks should therefore have more additional capabilities not captured within the task. This is operationalized by assuming that capabilities are equivalent to the number of neurons, which is a faulty assumption. As several reviewers note, the experiments do not adequately demonstrate the arguments of the paper, with even a validation of the core idea being absent.

Notes:
This is a really sloppy paper that basically assumes the conclusions and then doesn't test them. Definitely reject.

**Reviewer Concerns:**

see "Summary" text

**Reviewer Scores:**

see "Summary" text

---

### Decision · Program_Chairs · 2026-01-26

Reject